# Effect of Tension-Compression Asymmetry Response on the Bending of Prismatic Martensitic SMA Beams: Analytical and Experimental Study

**DOI:** 10.3390/ma14185415

**Published:** 2021-09-18

**Authors:** Alireza Ostadrahimi, Fathollah Taheri-Behrooz, Eunsoo Choi

**Affiliations:** 1School of Mechanical Engineering, Iran University of Science and Technology, Tehran 16846-13114, Iran; alireza_ostadrahimi@mecheng.iust.ac.ir or; 2Department of Civil Engineering, Hongik University, Seoul 04066, Korea

**Keywords:** analytical solution, shape memory alloy, tension-compression asymmetry, bending behavior, martensite beams

## Abstract

This paper aims to analytically derive bending equations, as well as semi-analytically predict the deflection of prismatic SMA beams in the martensite phase. To this end, we are required to employ a simplified one-dimensional parametric model considering asymmetric response in tension and compression for martensitic beams. The model takes into account the different material parameters in martensite twined and detwinned phases as well as elastic modulus depending on the progress of the detwinning process. In addition, the model considers the diverse slope of loading and unloading in martensite detwinned phases favored by tension and compression. To acquire general bending equations, we first solve the pure bending problem of a prismatic SMA beam. Three different phases are assumed in the unloading procedure and the effect of neutral fiber distance from the centerline is also considered during this stage. Then according to the pure bending solution and employing semi-analytical methods, general bending equations of an SMA beam are derived. Polynomial approximation functions are utilized to obtain the beam deflection–length relationship. To validate the attained analytical expressions, several three- and four-point bending tests were conducted for rectangular and circular SMA beams. Experimental data confirm the reasonable accuracy of the analytical results. This work may be envisaged to go deep enough in investigating the response of SMA beams under an arbitrary transverse loading and stress distribution during loading and unloading, as well as findings may be applicable to a good prediction of bending behavior.

## 1. Introduction

One of the smart class of materials distinguished by its two well-known behavior and a highly nonlinear stress–strain curve is shape memory alloy (SMA). Exhibiting pseudo-elasticity (PE) and shape memory effect (SME), as well as thermo-mechanically coupled responses have extensively opened up ample opportunities to utilize SMAs in different industrial sectors. SMAs can be used in numerous fields, including aerospace (e.g., jet engine, wing, Boeings variable geometry), biomedical (e.g., orthopedics, radiology, stents), robotics (e.g., walker, crawler, jumper), automotive (e.g., micro scanner system, side mirror actuator, tumble flaps actuator), and other consumer products [1,2,3].

Since in many fields of engineering applications, SMAs devices are exposed to mixed loading states, such as global or local bending, clarifying how SMA beam behaves under transverse loadings recently draws a considerable attention [4]. This great motivation leads to researchers coming up with various bending analysis via several experimental, numerical, and analytical studies in literatures. In terms of experimental and numerical works, several researchers [5,6,7,8,9,10,11,12,13,14,15,16,17] predicted bending stress distribution of SMA beams. Moreover, Refs. [18,19,20,21,22,23] considered asymmetric response in tension-compression and then compared their numerical results with experiments for both PE and SME beams. The results obtained from numerical calculations are significantly affected by secondary parameters including tolerance criteria, mesh size, number of load increments, and nonlinear geometry. These factors can lead to high computational expenses and time consumption as well as convergence difficulties [23].

On the other side, strong mathematical backing and high accuracy of analytical formulations persuade several attempts to study the behavior of SMA beams under the bending problem. In [23] by ignoring phase transformation hardening, they employed a piecewise linear constitutive equation to study the bending of a PE beam. A similar approach was used in [24,25,26,27] to obtain a close-form solution and moment–curvature relationship of the PE beams. The first authors proposed an analytical model to determine beam deflection and shear force in the general bending problem of both PE and SME beams, however, stress–strain response was assumed symmetric [28]. Moreover, the authors in [29] described twinning deformation effect of pre-strained SME beams subjected to the bending stresses. An exact solution for deflection of a PE cantilever beam loaded at the tip is derived in and bending stress of PE beams considering asymmetric response in tension-compression accurately calculated [4,30]. Several theoretical works were also studied for bending of laminated composite PE beams considering the constitutive equations with different material behavior in tension and compression [31,32,33]. More recently, assuming the effect of tension–compression asymmetry, a constitutive model to predict the response of PE and SME beams under a three-point bending test and pure bending is proposed in [34,35], respectively.

Among the existing theoretical models for bending problem, most cases are dedicated to analyzing the flexural response of the PE beams. But experimental data reported in [36] (Figure 1) also proves the significant importance of asymmetric behavior in tension and compression for SMA at low temperatures. Although some researchers [21,37,38,39,40] have proposed appropriate phenomenological models which are capable to describe non-symmetric response in tension and compression, investigation of the bending behavior of SME beams which analytically addresses asymmetric response of martensite phase in tension and compression has not been reported yet.

Employing the existing tension–compression asymmetric models seem well-adapted for numerical simulations; however, they could be complex enough to derive analytical solutions for predicting beam deflection under transverse loading. In addition, those with close-form potential solutions have not considered any important secondary effects for bending problem, such as diverse hardening in detwinning zones as well as different elastic modulus during loading and unloading in detwinned martensite zones in tension and compression. Thus, in this paper based on the experimental data reported in [36], a parametric model is proposed to enable a precise description of asymmetric stress–strain behavior in loading and unloading. Then the bending behavior of SMA beams at low temperatures with both analytical and experimental approaches is studied. Employing the parametric model and according to Euler-Bernoulli beam theory, critical curvatures upon initiation of each zone in tension and compression are obtained. Moreover, moment–curvature relationships of an SME beam subjected to pure bending problem are derived. Based on the provided solution for pure bending, we expand our work to acquire governing equations of an SMA beam subjected to an arbitrary transverse loading. Furthermore, the deflection along the beam length is obtained for several beams with different cross-sections. To validate the results of this work, we have experimentally conducted several three- and four-point bending tests for rectangular and circular SMA beams. This study is structured as follows. Section 2 represents goals and objectives of this study. In Section 3, a parametric model is proposed to describe the asymmetric behavior of martensite SMA in tension and compression. The relationship between moment and curvature as well as the distance of neutral fiber from the centerline during loading is discussed in Section 4. Then in Section 5, considering the reverse detwinning process during unloading, we capture the relationship between stress and height of the beam which results in twelve different unloading states. Using semi-analytical method, pure bending equations are developed for an SMA beam under transverse loading in Section 6, which addresses the beam deflection along the length. Section 7 offers the experimental tests conducted for tensile, compressive, and bending as for finding out mechanical characterizations of SMA as well as differential scanning calorimetry test to determine phase transformation temperatures. Several bending cases are solved in Section 8, and then analytical results are compared with experimental data. In the last section, we finally provide a summary and draw conclusions.

## 2. Goals and Objectives of the Study

As stated in Section 1, most of the analytical studies on the bending of SMA beams are dedicated to superelastic features as well as austenite phase, however, martensitic beams attracted less attention and only have considered pure bending problem with symmetric material behavior. For this purpose, considering asymmetric tension and compression response with careful attention to all asymmetric behavior presented in Section 4 and Section 5, general bending of SME beam under transverse loading is derived and the relationship between beam deflection–length and moment–curvature during loading and unloading for rectangular and circular beams is obtained using polynomial approximation function and then compared with the experimental data as well as numerical results. Proposing explicit expressions for bending of SME beam empowers the designer to simply determine beam deflection and stress distribution along the SME beam when applied arbitrary transverse loading in general bending problem.

## 3. Parametric Model to Describe Asymmetric Tension-Compression Behavior in Martensite Phase

Motivated by one-dimensional SMA model proposed by Auricchio et al. [37] at low temperature, we present a parametric model for martensitic SMA beams to describe material behavior with asymmetric tension–compression response. This simplified one-dimensional parametric model enables us to derive bending behavior and predict the SMA beam deflection analytically. Material parameters in tension and compression are denoted with superscripts + and −, respectively. For more clarity and conciseness, Figure 2 only depicts the parameters in tension. For SME stress–strain curve in literatures, three different zones of twin (elastic), variant reorientation (VR-), and detwinned (D-) zones are generally defined. However, in this study to model the stress–strain curve more accurately, we have introduced and added two transient zones, namely T1- (elastic to VR-) zone and T2- (VR- to D-) zone. We express general one-dimensional constitutive relation as
(1)$$\sigma =E(\varepsilon -e^{tr})$$
where E and $e^{tr}$ are the Young modulus and transformation strain. Table 1 presents the material parameters related to each zone during loading in tension and compression. $E_{Tj}^{\pm }$ (*j* = 1, 2), $E_{VR}^{\pm }$, $E_{D}^{\pm }$ as well as $\varepsilon_{Tj}^{\pm }$, $\varepsilon_{VR}^{\pm }$, and $\varepsilon_{D}^{\pm }$ denote modulus of elasticity and transformation strain in T*j*-, VR-, and D-zone, respectively. Moreover, the amounts of elastic moduli and transformation strains for each zone are determined and listed in Table 2. During loading, the stress and strain may exceed the maximum detwinning stress (strain) so that some fibers may be located in D zone. The stress and strain in this region are specified by $\sigma_{det}^{\pm }$ and $\varepsilon_{det}^{\pm }$. Moreover, $\varepsilon_{L}^{\pm }$ denotes the remained strain in the fiber upon unloading and it has the same value for all fibers in D zone.

In this model, to describe stress–strain behavior during unloading, we assume the unloading process (Figure 2) may consist of three parts as:Elastic unloading (U1);Stress reduction due to the reverse transient zone (U2);Stress reduction due to the reverse variant reorientation (U3).

Moreover, based on the experimental data reported in [36], unloading slope during phase transformation could be affected by the loading stress. Even in D-zone, the slope of unloading may not be coincident with the loading slope which is not considered in the proposed model by Auricchio et al. [37]. Accordingly, elastic modulus during unloading from $z_{i}$ fiber ($E_{U1,i}^{\pm }$, $E_{U2,i}^{\pm }$ and $E_{U3,i}^{\pm }$) can be modeled as

**VR-zone:**(2)$$(\begin{array}{l} E_{U1,i}^{\pm }=\frac{(\sigma_{i}^{\pm }-\sigma_{S}^{\pm })E_{U1,D}^{\pm }+(\sigma_{M}^{\pm }-\sigma_{i}^{\pm })E}{\sigma_{M}^{\pm }-\sigma_{S}^{\pm }}\;\;\;\;\;\;\;\;\;\;\;\;\;\;\;\;\;\;\;\;\;\;\;\;\;(a)\\ E_{U2,i}^{\pm }=\frac{E_{U1,i}^{\pm }}{E_{U1,D}^{\pm }}E_{U2,D}^{\pm }\;\;\;\;\;\;\;\;\;\;\;\;\;\;\;\;\;\;\;\;\;\;\;\;\;\;\;\;\;\;\;\;\;\;\;\;\;\;\;\;\;\;\;\;(b)\\ E_{U3,i}^{\pm }=\frac{\sigma_{R}^{\pm }-\sigma_{F}^{\mp }}{\varepsilon_{R,i}^{\pm }-\varepsilon_{F}^{\mp }}\;\;\;\;\;\;\;\;\;\;\;\;\;\;\;\;\;\;\;\;\;\;\;\;\;\;\;\;\;\;\;\;\;\;\;\;\;\;\;\;\;\;\;\;\;\;\;\;\;\;\;(c) \end{array}$$ where $E_{U1,D}^{\pm }$, $E_{U2,D}^{\pm }$ and $E_{U3,D}^{\pm }$ represent elastic moduli related to each part of unloading (U1, U2, and U3) in D-zone as:

**D-zone:**(3)$$(\begin{array}{l} E_{U1,D}^{\pm }=\frac{\sigma_{det}^{\pm }}{\varepsilon_{det}^{\pm }-\varepsilon_{L}^{\pm }}\;\;\;\;\;\;\;\;\;\;\;\;\;\;\;\;\;\;\;\;\;\;\;\;\;\;\;\;\;\;\;(a)\\ E_{U2,D}^{\pm }=\frac{-\sigma_{R}^{\pm }}{\varepsilon_{L}^{\pm }-\varepsilon_{R,i}^{\pm }}\;\;\;\;\;\;\;\;\;\;\;\;\;\;\;\;\;\;\;\;\;\;\;\;\;\;\;\;\;\;\;(b) \end{array}$$$E_{U3,D}^{\pm }$ can be obtained from Equation (2c). Moreover, $\sigma_{R}^{\pm }$ and $\varepsilon_{R,i}^{\pm }$ denote reverse VR start stress and strain, respectively. $\sigma_{R}^{\pm }$ takes a constant value and for simplicity we consider it equal to $\sigma_{P}^{\mp }$. While $\varepsilon_{R,i}^{\pm }$ could be a function of loading stress ($\sigma_{i}^{\pm }$) and strain ($\varepsilon_{i}^{\pm }$) which may be determined as:(4)$$\varepsilon_{R,i}^{\pm }=\varepsilon_{i}^{\pm }-\frac{\sigma_{i}^{\pm }-\sigma_{R}^{\pm }}{E_{U1,i}^{\pm }}$$

## 4. Moment–Curvature Relationship in Loading

Using stress–strain curve represented in Figure 2 and based on Euler–Bernoulli beam theory in which beam cross-section remains plane during the elastic-plastic deformation, we derive an explicit relationship between bending moment and curvature during the loading procedure. Stress–height diagrams corresponding to each level of bending stress are schematically illustrated in Figure 3. In the most general cases, as the outmost fibers of tensile and compressive sides are located in detwinned zones, we can divide the cross-section into nine diverse regions as indicated in Figure 3e.

Due to the material parameters and asymmetric behavior of SMAs, the neutral axis of the beam will not be coincident with the centroid and may shift down or up concerning the centerline. Considering an arbitrary symmetric cross-section about *y*- and *z*-axes with maximum height *h* and thickness *t* (Figure 4), the moment–curvature relationship for the most general case is studied.

Reminding Euler beam theory, the strain (ε) and curvature (κ) relationship can be expressed as:(5)ε=−κ(z−d)
where *d* represents the distance of neutral fiber from the centerline (*z = 0*). Due to the general asymmetric behavior of SMAs (Figure 1), we assume that $\sigma_{P}^{-}$ is higher than $\sigma_{P}^{+}$ (however, for $\sigma_{P}^{+}<\sigma_{P}^{-}$ the same process could be followed). Considering Figure 3a in which no detwinning process can occur, stress distribution still would be symmetric; thus, *d* is zero. Reaching stress of the outmost fiber maximum proportional stress in tension, the corresponding curvature ($\kappa_{E}^{+}$) may be obtained as:(6)$$\kappa_{E}^{+}=\frac{2\sigma_{P}^{+}}{Eh}$$

Exceeding $\kappa_{E}^{+}$ may initiate T1-zone in tension, however, additional loading stress is required to start this process in the compressive region (Figure 3b); thus, we have
(7)$$\kappa_{E}^{-}=\frac{\sigma_{P}^{-}}{E(\frac{h}{2}-d)}$$

The unknown parameter of d can be expressed in terms of κ and $z_{P}^{+}$ by satisfying the equilibrium of force (∫σdA=0), which reads
(8)$$\int_{-h/2}^{z_{P}^{+}}E_{T1}^{+}(\varepsilon -\varepsilon_{T1}^{+})\;dA+\int_{z_{P}^{+}}^{h/2}E\varepsilon \;dA=0$$
where *A* represents beam cross section. When the stress in tensile and compressive side achieves $\sigma_{S}^{\pm }$, related curvatures to these stresses, respectively, yield
(9)$$\kappa_{T1}^{\pm }=\frac{2}{h\pm 2d}\;\left(\frac{\sigma_{S}^{\pm }}{E_{T1}^{\pm }}+\varepsilon_{T1}^{\pm }\right)$$

Then according to Figure 3c, the updated d can be obtained as:(10)$$\int_{-h/2}^{z_{P}^{+}}E_{T1}^{+}(\varepsilon -\varepsilon_{T1}^{+})\;dA+\int_{z_{P}^{+}}^{z_{P}^{-}}E\varepsilon \;dA+\int_{z_{P}^{-}}^{h/2}E_{T1}^{-}(\varepsilon -\varepsilon_{T1}^{-})\;dA=0$$

In Equations (8) and (10), $z_{P}^{\pm }$ are the distance between the centerline and the boundary of elastic and T1-zone in tension and compression; thus, using Equation (5) and Table 1, they can be obtained as
(11)$$z_{P}^{\pm }=d-\frac{\sigma_{P}^{\pm }}{E\kappa }$$

Further increase in applied load attributes more fibers in tension and/or compression experience VR-zone. Regarding Figure 1, finish stress of VR-zone in tension could be greater than compression ($\sigma_{F}^{-}<\sigma_{F}^{+}$), however, for $\sigma_{F}^{+}<\sigma_{F}^{-}$ we can follow a similar procedure. Considering Figure 3e, and using Table 1, maximum curvature of VR-zone in tension and compression ($\kappa_{VR}^{\pm }$) may be written as:(12)$$\kappa_{VR}^{\pm }=\frac{2}{h\pm 2d}\;\left(\frac{\sigma_{F}^{\pm }}{E_{VR}^{\pm }}+\varepsilon_{VR}^{\pm }\right)$$

Similar to Equation (10), we must refresh d in terms of curvature as
(13)$$\begin{array}{l} \int_{-h/2}^{z_{S}^{+}}E_{VR}^{+}(\varepsilon -\varepsilon_{VR}^{+})\;dA+\int_{z_{S}^{+}}^{z_{P}^{+}}E_{T1}^{+}(\varepsilon -\varepsilon_{T1}^{+})\;dA+\\ \int_{z_{P}^{+}}^{z_{P}^{-}}E\varepsilon \;dA+\int_{z_{P}^{-}}^{z_{S}^{-}}E_{T1}^{-}(\varepsilon -\varepsilon_{T1}^{-})\;\;dA+\int_{z_{S}^{-}}^{h/2}E_{VR}^{-}(\varepsilon -\varepsilon_{VR}^{-})\;\;dA=0 \end{array}$$
where $z_{S}^{\pm }$ expresses the distance from the boundary of VR- and T2-zone in tension and compression. Thus, using Equation (5) and Table 1 they may be written as
(14)$$z_{S}^{\pm }=d-\frac{1}{\kappa }\left(\frac{\sigma_{S}^{\pm }}{E_{T1}^{\pm }}+\varepsilon_{T1}^{\pm }\right)$$

Finally, achieving stresses in the outmost fibers in tension and compression $\sigma_{M}^{\pm }$ (Figure 3g) causes the second transient zone (T2-zone) to begin developing in each side; thus, corresponding curvatures ($\kappa_{T2}^{\pm }$) are derived as
(15)$$\kappa_{T2}^{\pm }=\frac{2}{h+2d}\;\left(\frac{\sigma_{M}^{\pm }}{E_{T2}^{\pm }}-\varepsilon_{T2}^{\pm }\right)$$

According to Figure 3i, and then employing $\int_{A}\sigma zdA$ applied bending moment for the most general loading case is obtained as
(16)$$\begin{array}{l} M_{loading}=\int_{-h/2}^{z_{M}^{+}}E_{D}^{+}(\varepsilon -\varepsilon_{D}^{+})z\;dA+\int_{z_{M}^{+}}^{z_{F}^{+}}E_{T2}^{+}(\varepsilon -\varepsilon_{T2}^{+})z\;dA+\\ \int_{z_{F}^{+}}^{z_{S}^{+}}E_{VR}^{+}(\varepsilon -\varepsilon_{VR}^{+})\;zdA+\int_{z_{S}^{+}}^{z_{P}^{+}}E_{T1}^{-}(\varepsilon -\varepsilon_{T1}^{-})z\;\;dA+\int_{z_{P}^{+}}^{z_{P}^{-}}E\varepsilon z\;\;dA+\int_{z_{P}^{-}}^{z_{S}^{-}}E_{T1}^{+}(\varepsilon -\varepsilon_{T1}^{+})\;zdA+\\ \int_{z_{S}^{-}}^{z_{F}^{-}}E_{VR}^{-}(\varepsilon -\varepsilon_{VR}^{-})\;zdA+\int_{z_{F}^{-}}^{z_{M}^{-}}E_{T2}^{-}(\varepsilon -\varepsilon_{T2}^{-})z\;\;dA+\int_{z_{M}^{-}}^{h/2}\;E_{D}^{-}(\varepsilon -\varepsilon_{D}^{-})z\;dA \end{array}$$

Furthermore, force equilibrium must be enforced to find the new amount of *d* as
(17)$$\begin{array}{l} \int_{-h/2}^{z_{M}^{+}}E_{D}^{+}(\varepsilon -\varepsilon_{D}^{+})\;dA+\int_{z_{M}^{+}}^{z_{F}^{+}}E_{T2}^{+}(\varepsilon -\varepsilon_{T2}^{+})\;dA+\int_{z_{F}^{+}}^{z_{S}^{+}}E_{VR}^{+}(\varepsilon -\varepsilon_{VR}^{+})\;dA+\int_{z_{S}^{+}}^{z_{P}^{+}}E_{T1}^{-}(\varepsilon -\varepsilon_{T1}^{-})\;\;dA+\\ \int_{z_{P}^{+}}^{z_{P}^{-}}E\varepsilon \;\;dA+\int_{z_{P}^{-}}^{z_{S}^{-}}E_{T1}^{+}(\varepsilon -\varepsilon_{T1}^{+})\;dA+\int_{z_{S}^{-}}^{z_{F}^{-}}E_{VR}^{-}(\varepsilon -\varepsilon_{VR}^{-})\;dA+\int_{z_{F}^{-}}^{z_{M}^{-}}E_{T2}^{-}(\varepsilon -\varepsilon_{T2}^{-})\;\;dA+\\ \int_{z_{M}^{-}}^{h/2}\;E_{D}^{-}(\varepsilon -\varepsilon_{D}^{-})\;dA=0 \end{array}$$

To obtain the distance of boundary of T2- and D-zone in tension and compression from the centerline ($z_{F}^{\pm }$), we first attain the height of VR-zones in tension and compression considering variant reorientation start and finish strains which could be expressed as
(18)$$z_{F}^{\pm }-z_{S}^{\pm }=\frac{\varepsilon_{F}^{\pm }-\varepsilon_{S}^{\pm }}{\kappa }$$

Thus, using Equations (14) and (18), $z_{F}^{\pm }$ may be determined as
(19)$$z_{F}^{\pm }=d-\frac{1}{\kappa }\left(\varepsilon_{S}^{\pm }-\varepsilon_{F}^{\pm }+\varepsilon_{T1}^{\pm }+\frac{\sigma_{S}^{\pm }}{E_{T1}^{\pm }}\right)$$

Finally, considering half of the beam height, $z_{M}^{\pm }$ can be easily obtained as
(20)$$z_{M}^{\pm }=\frac{h}{2}-\left(z_{P}^{\pm }+z_{S}^{\pm }+z_{F}^{\pm }\right)$$

## 5. Moment–Curvature Relationship in Unloading

While in the bending problem of SME beams, most of the researchers have considered stress reduction during unloading purely elastic, it has been proven that as elastic strains are recovered, reverse detwinning process may then occur [29]. However, in this study to increase the accuracy of the model, three unloading phases introduced in Section 2 are considered, so that fibers after elastic unloading go through the reverse transient zone as well as experience reverse variant reorientation.

For considering symmetric material behavior, it is assumed that in the unloading procedure, neutral fiber is located at the centroid. However, in the reality after unloading, neutral fiber may still gradually move up or down to satisfy force equilibrium. Therefore, to involve this effect in the present work, the stress distribution is considered asymmetric so that the distance of neutral fiber from the centerline upon unloading is represented by $d_{u}$.

Unloading behavior is not only affected by the loading curvature but also by the asymmetric material behavior; the location of neutral fiber plays a significant role in capturing different forms of stress–height diagram during unloading. Generally, twelve unloading cases might be expected (Figure 5a–l). To elucidate the unloading procedure, stress–height diagram (Figure 5) and stress–strain curve (Figure 6) are numbered and individually described as below:All fibers are in the elastic zone (Figure 5a); thus, the assumption of elastic unloading with the same modulus of elasticity (*E*) is reliable.In Figure 5b,c, respectively, some tensile ($-h/2<z<z_{2}$) and compressive ($z_{4}<z<h/2$) fibers experience T1-zone while unloading is still symmetric with the slope of *E*.Initiation of VR-zone is favored by tension and compression (Figure 5d,e) so that upon unloading the modulus of elasticity is a function of loading stress; thus, each fiber has its own $E_{U1,i}^{\pm }$.Figure 5f describes a special position of $z_{S}^{+}$, so that upon unloading it is coincident with $d_{u}$. After this state, as unloading occurs some fibers reload with the slope of $E_{VR}^{+}$.Some fibers in VR-zone are favored by tension, in addition to elastic unloading, undergo stress reduction due to the reverse T-zone and reverse VR-zone $E_{U2,i}^{+}$ and $E_{U3,i}^{+}$ slopes, respectively (Figure 5g).According to Figure 5h, some fibers in VR-zone favored by compression, ($z_{8}<z<h/2$) unload with moduli of $E_{U2,i}^{-}$ and $E_{U3,i}^{-}$ in reverse T- and VR-zone, respectively.Once the compressive and tensile stresses exceed $\sigma_{F}^{\pm }$, it causes some fibers to enter into T2-zone and consequently stress releases similar to case 6. Moreover, we anticipate some fibers sit in single variant zones follow the same trend, however, in this case, slope of unloading for reverse T- and VR-zones are $E_{U2,D}^{-}$ and $E_{U3,D}^{-}$, respectively (Figure 5i,j).Further proceeding into a single variant favored by tension and/or compression (Figure 5k,l), purely elastic unloading reappears during stress releasing.

Figure 5 indicates the most general case where all unloading states can simultaneously occur. To set out in detail, Figure 6 clarifies how unloading process can act as below:

Elastic unloading for paths 3–7 and 14–15 in tension, as well as 1–4, 4–6, 6–8, and 13–16 in compression.

Reloading for points 1–2, 2–5, and 5–3 in tension.After elastic unloading fibers between points 7–10 and 10–12 in tension as well as 8–9 and 9–11 in compression have increasing trend in reverse T- and VR-zones.Fibers between 12–14 and 11–13 in tension and compression, respectively, suffer from a decreasing trend in reverse T- and VR-zones.

According to the above discussion, released stresses for $z_{i}$ can be decomposed into elastic unloading ($\Delta \sigma_{i}^{E}$), stress reduction due to the reverse T- ($\Delta \sigma_{i}^{T}$) and VR-zone ($\Delta \sigma_{i}^{VR}$), as
(21)$$\Delta \sigma_{i}=\Delta \sigma_{i}^{E}+\Delta \sigma_{i}^{T}+\Delta \sigma_{i}^{VR}$$

Thus, bending moment during unloading may be expressed as
(22)$$\begin{array}{l} M_{U}=\int_{-h/2}^{z_{14}}\Delta \sigma_{i}^{E}(z-d_{u})\;dA+\int_{z_{14}}^{z_{10}}(\Delta \sigma_{i}^{E}+\Delta \sigma_{i}^{T}+\Delta \sigma_{i}^{VR})(z-d_{u})\;dA+\int_{z_{10}}^{z_{7}}(\Delta \sigma_{i}^{E}+\Delta \sigma_{i}^{T})(z-d_{u})\;dA+\\ \int_{z_{7}}^{z_{3}}\Delta \sigma_{i}^{E}(z-d_{u})\;dA+\int_{z_{3}}^{z_{2}}(\Delta \sigma_{i}^{T}+\Delta \sigma_{i}^{VR})(z-d_{u})\;dA+\int_{z_{2}}^{z_{8}}\Delta \sigma_{i}^{E}(z-d_{u})\;dA+\\ \int_{z_{8}}^{z_{9}}(\Delta \sigma_{i}^{E}+\Delta \sigma_{i}^{T})(z-d_{u})\;dA+\int_{z_{9}}^{z_{13}}(\Delta \sigma_{i}^{E}+\Delta \sigma_{i}^{T}+\Delta \sigma_{i}^{VR})(z-d_{u})\;dA+\int_{z_{13}}^{h/2}\Delta \sigma_{i}^{E}(z-d_{u})\;dA \end{array}$$

As mentioned, $\sigma_{i}^{+}$ and $\sigma_{i}^{-}$ are the loading stress in tension and compression, respectively. To determine bending moment, it is required to obtain the unknown parameters of stress ($\Delta \sigma_{i}^{e}$, $\Delta \sigma_{i}^{T}$ and $\Delta \sigma_{i}^{VR}$) and height ($z_{3}$ or $d_{u}$, $z_{7}$, $z_{8}$, $z_{13}$ and $z_{14}$), however, other parameters of height have just been specified from loading section. Due to the complexity of behavior, unknown parameters of stress and height may be initially derived in terms of $d_{u}$ (or $z_{3}$) and curvature reduction (Δκ), then to obtain $M_{U}$, we employ force equilibrium equation to express $d_{u}$ in terms of Δκ. Unknown stress parameters can be related to Δκ using strain expressions; thus, considering Equation (5) and strain-intercept during unloading we may write as
(23)$$\Delta \varepsilon_{i}=-\Delta \kappa \left(z_{i}-d_{u}\right)$$

According to Equation (21), we decompose strain reduction for $z_{i}$, into elastic strain reduction ($\Delta \varepsilon_{i}^{e}$), strain reduction due to the reverse transient ($\Delta \varepsilon_{i}^{T}$) and variant reorientation zones ($\Delta \varepsilon_{i}^{VR}$) parts, as
(24)$$\Delta \varepsilon_{i}=\Delta \varepsilon_{i}^{e}+\Delta \varepsilon_{i}^{T}+\Delta \varepsilon_{i}^{VR}$$

Regarding the process of unloading described in Figure 6, we express $\Delta \varepsilon_{i}^{e}$ as
(25)$$\Delta \varepsilon_{i}^{e}=\left\{ \begin{array}{l} \frac{\sigma_{i}^{+}}{E_{U1,i}^{+}}\;\;\;\;\;\;\;z_{14}\leq z_{i}\leq z_{7}\\ 0\;\;\;\;\;\;\;\;\;\;\;\;\;\;\;z_{3}\leq z\leq z_{2}\\ \frac{\sigma_{i}^{-}}{E_{U1,i}^{-}}\;\;\;\;\;\;\;z_{8}\leq z_{i}\leq z_{13}\\ \Delta \varepsilon_{i}^{}\;\;\;\;\;\;\;\;\;otherwise \end{array}\right.$$

Using Table 1 for each region and then substituting Equations (5) and (25) into Equation (24), as well as employing $\Delta \sigma_{i}^{T}=E_{U2}^{\pm }\Delta \varepsilon_{i}^{T}$ and $\Delta \sigma_{i}^{VR}=E_{U3,i}^{\pm }\Delta \varepsilon_{i}^{VR}$, we may finally write $\Delta \sigma_{i}^{VR}$ as
(26)$$\begin{array}{l} \Delta \sigma_{i}^{VR}=\left\{ \begin{array}{l} E_{U3,i}^{\pm }\left(-\Delta \kappa (z_{i}-d_{u})+\frac{E_{VR}^{\pm }(\kappa z_{i}+\varepsilon_{VR}^{\pm }-\kappa d)}{E_{U1,i}^{\pm }}+\frac{\sigma_{R}^{\pm }}{E_{U2,i}^{\pm }}\right)\;\;\;\;z_{10}\leq z_{i}\leq z_{7},\;\;z_{8}\leq z_{i}\leq z_{9}\\ E_{U3,i}^{\pm }\left(-\Delta \kappa (z_{i}-d_{u})+\frac{E_{T2}^{\pm }(\kappa z_{i}+\varepsilon_{T2}^{\pm }-\kappa d)}{E_{U1,i}^{\pm }}+\frac{\sigma_{R}^{\pm }}{E_{U2,i}^{\pm }}\right)\;\;\;\;z_{12}\leq z_{i}\leq z_{10}\;,\;\;z_{9}\leq z_{i}\leq z_{11}\\ E_{U3,D}^{\pm }\left(-\Delta \kappa (z_{i}-d_{u})+\frac{E_{D}^{\pm }(\kappa z_{i}+\varepsilon_{D}^{\pm }-\kappa d)}{E_{U1,D}^{\pm }}+\frac{\sigma_{R}^{\pm }}{E_{U2,D}^{\pm }}\right)\;\;\;\;z_{14}\leq z_{i}\leq z_{12}\;,\;\;z_{11}\leq z_{i}\leq z_{13} \end{array}\right. \end{array}$$

Furthermore, reloading for points 2 to 3 attributes to stress increase in VR-zone; thus, using Equation (23) it yields
(27)$$\Delta \sigma_{i}^{}=-E_{VR}^{+}\Delta \kappa \left(z_{i}-d_{u}\right)$$

To derive $z_{7}$, it should be considered that point 7 is exposed to experience reverse variant reorientation during unloading, so that at this moment $\Delta \varepsilon_{7}^{VR}$ is still zero and $\Delta \sigma_{i}^{T}$ reach $-\sigma_{R}^{+}$; thus, using Equations (21) and (24) we have
(28)$$\left(\Delta \varepsilon_{7}-\Delta \varepsilon_{7}^{E}\right)E_{U2,7}^{+}=-\sigma_{R}^{+}$$

Inserting the constitutive equation of VR-zone (Table 1) and Equations (2a) into (2b), $E_{U2,7}^{+}$ can be determined in terms of $z_{7}$. Then substituting Equations (23) and (25) into (28) and after some mathematical manipulations, we achieve a polynomial expression in terms of $z_{7}$ as
(29)$$A_{1}z_{7}^{2}+A_{2}z_{7}+A_{3}=0$$
where $A_{1}$, $A_{2}$ and $A_{3}$ can be expressed as
(30)$$\begin{array}{l} A_{1}=\frac{\kappa \Delta \kappa E_{VR}^{+}E_{U2,D}^{+}}{\sigma_{M}^{+}-\sigma_{S}^{+}}\left(\frac{E}{E_{U1,D}^{+}}-1\right)\;\;\;\;\;\;\;\;\;\;\;\;\;\;\;\;\;\;\;\;\;\;\;\;\;\;\;\;\;\;\;\;\;\;\;\;\;\;\;\;\;\;\;\;\;\;\;\;\;\;\;\;\;\;\;\;\;\;\;\;\;\;\;\;\;\;\;\;\;\;\;\;\;\;\;\;\;\;\;\;\;\;\;\;\;\;\;\;\;\;\;\;\;(a)\\ A_{2}=\kappa E_{VR}^{+}\frac{E_{U2,D}^{+}}{E_{U1,D}^{+}}-\frac{A_{1}}{\kappa }\left(\varepsilon_{VR}^{+}+\frac{\sigma_{M}^{+}+\sigma_{S}^{+}}{E_{VR}^{+}}\right)+A_{1}\left(d+d_{u}\right)\;\;\;\;\;\;\;\;\;\;\;\;\;\;\;\;\;\;\;\;\;\;\;\;\;\;\;\;\;\;\;\;\;\;\;\;\;\;\;\;\;\;\;\;\;\;\;\;\;\;\;\;\;\;(b)\\ A_{3}=A_{1}d_{u}\left(\frac{\varepsilon_{VR}^{+}}{\kappa }-d\right)+E_{VR}^{+}\frac{E_{U2,D}^{+}}{E_{U1,D}^{+}}\left(\varepsilon_{VR}^{+}-\kappa d\right)+\frac{E_{U2,D}^{+}\Delta \kappa d_{u}}{\sigma_{M}^{+}-\sigma_{S}^{+}}\left(\frac{E}{E_{U1,D}^{+}}\sigma_{M}^{+}-\sigma_{S}^{+}\right)+\sigma_{R}^{+}\;\;\;\;\;\;\;\;(c) \end{array}$$

The fiber at point 14 during unloading has arrived at zero stress level ($\sigma_{14}-\Delta \sigma_{14}=0$); thus, only elastic portion of Equation (21) exists. Employing $\Delta \sigma_{14}=E_{U1,D}^{+}\Delta \varepsilon_{14}$, Equation (5) as well as considering constitutive relation of D-zone in tension (Table 1), $z_{14}$ may be given as
(31)$$z_{14}=\frac{E_{U1,D}^{+}\Delta \kappa d_{u}-E_{D}^{+}\left(\kappa d-\varepsilon_{L}^{+}\right)}{\Delta \kappa E_{U1,D}^{+}-\kappa E_{D}^{+}}$$

In a similar approach, we can determine $z_{8}$ and $z_{13}$, however, corresponding parameters to compressive side in Equations (29)–(31) must be considered. Finally, balancing the equilibrium of force in Figure 5i, leads to derivation of $d_{u}$ in terms of Δκ as
(32)$$\begin{array}{l} \int_{-h/2}^{z_{14}}\Delta \sigma_{i}^{E}\;dA+\int_{z_{14}}^{z_{10}}(\Delta \sigma_{i}^{E}+\Delta \sigma_{i}^{T}+\Delta \sigma_{i}^{VR})\;dA+\int_{z_{10}}^{z_{7}}(\Delta \sigma_{i}^{E}+\Delta \sigma_{i}^{T})dA+\int_{z_{7}}^{z_{3}}\Delta \sigma_{i}^{E}\;dA+\\ \int_{z_{3}}^{z_{2}}(\Delta \sigma_{i}^{T}+\Delta \sigma_{i}^{VR})\;dA+\int_{z_{2}}^{z_{8}}\Delta \sigma_{i}^{E}\;dA+\int_{z_{8}}^{z_{9}}(\Delta \sigma_{i}^{E}+\Delta \sigma_{i}^{T})\;dA+\int_{z_{9}}^{z_{13}}(\Delta \sigma_{i}^{E}+\Delta \sigma_{i}^{T}+\Delta \sigma_{i}^{VR})\;dA+\int_{z_{13}}^{h/2}\Delta \sigma_{i}^{E}\;dA \end{array}$$

All unknown parameters thus far have been just expressed in terms of curvature reduction which may be calculated by fulfilling $M_{L}^{}+M_{U}^{}=0$.

## 6. Expanding Pure Bending Approach to Determine Beam Deflection in General Bending

Ostadrahimi et al. in [28] proposed an analytical solution based on the pure bending problem to achieve beam deflection. As for bending moment depends on the beam length, each cross-section represents a pure bending problem. Accordingly, for a beam with an arbitrary transverse loading, we initially derive bending moment for a cross-section. Employing polynomial approximation function, curvature may be subsequently expressed in terms of the SMA beam length. Considering a beam under transverse loading (Figure 7), it is expected to create nine types of cross-sections based on the stress–strain curve. However, for simplicity we consider elastic, VR- and D-zones in tension and compression, thus five types of cross-sections can be generated.

All fiber in the elastic zone.Some fibers are in VR-zone favored by tension (compression).VR-zones in both tension and compression sides have occurred.Some fibers are in D-zone favored by tension (compression).On both sides, D-zone can happen.

Moreover, $l_{i}$ (*i =* 1–5) indicates the corresponding length to each type of cross-sections, so that they can be derived when the bending moment is determined. As small deflection is assumed, we replace $\kappa (x)=d^{2}w/dx^{2}$ in Equation (22), leading to a nonlinear differential equation as a function of deflection. To solve this differential equation via a semi-analytical approach, a polynomial approximation function is employed to propose the beam curvature as
(33)$$\kappa (x)=a_{n}x^{n}+a_{n-1}x^{n-1}+\ldots +a_{1}x+a_{0}$$
where n denotes the polynomial maximum degree. To obtain $a_{0}$ to $a_{n}$, we substitute Equation (33) into Equation (22) and then equate coefficients of $x^{0}$ to $x^{n}$ on both sides. Integrating Equation (33), deflection along the length (x) is finally represented as:(34)$$w(x)=\int_{0}^{x}\left(\int_{0}^{x}\kappa (x)\;dx\right)\;dx+C_{1}x+C_{2}$$

Imposing appropriate boundary conditions, we may determine integration constants of $C_{1}$ and $C_{2}$. 

## 7. Experimental Tests

Most of the previous experimental studies on SMA bending problem are focused on the behavior of superelastic beams while in this work, several experimental bending tests with different cross-sections have been conducted to broaden the physical understanding of martensitic beams under bending behavior and compare experimental data with the proposed analytical solution.

To find out material properties and mechanical characterizations of the NiTi alloy (Ni: 53 wt%, Ti: 47 wt%) martensitic beam, we have initially measured transformation temperatures through differential scanning calorimetry (DSC) test, indicated in Figure 8; M_S_, M_F_, A_S_, and A_F_ are, respectively, martensite and austenite start and finish temperatures.

According to the ASTM E8/E8M-16AE1 and E9-19 (2016), several standard samples are fabricated, and then using the Universal Test Machine (UTM), tensile and compressive tests are carried out to determine the stress–strain behavior (Figure 9a,b). Furthermore, three prismatic beams with circular (diameter 20 mm and length 300 mm) and two rectangular cross-sections (height 10 mm, thickness 14 mm and length 100 mm) are subjected to three- and four-points bending tests to measure the force–deflection relationship at different conditions (Figure 9c,d). All experiments have been performed under displacement control and at the speed of 1 mm/min. Note that, the manufacturer, model, and country of test device, respectively, for DSC are NETZSCH, LFA 467, Selb, Germany, and for tensile, compressive, and bending tests is MTS, 500 kN-370.50, Eden Prairie, MN, USA.

## 8. Experimental Data and Numerical Results

Several case studies are investigated in this section for martensitic SMA beams with rectangular and circular cross-sections. To this end, we have derived pure bending equations for martensitic beams and presented moment–curvature and force–displacement relationships under diverse loading conditions. Moreover, general bending equations based on the pure bending problem have been obtained by considering a cantilever beam. To validate the current work, we initially compared the results of the pure bending problem with the finite element method using ABAQUS software 6.14, Paris, France in which symmetric material behavior is assumed. Then the analytical results for martensitic beams while having different responses in tension and compression are compared with experimental data from three- and four-point bending tests.

**Example1:** Pure bending problem with a rectangular cross-section.

A rectangular beam of with *t*, height *h*, and length *l* under pure bending condition has been considered in Figure 10. Using the analytical solution in Section 3 and Section 4, Table 3 presents loading and unloading force and moment–curvature relationships for a rectangular martensite SMA beam.

In this example, we first consider symmetric material properties for SMA beam to compare the results of current work with the results of three-dimensional SMA continuum elements [3]. To simulate symmetric material properties by the finite element method, a user-defined material subroutine (UMAT) has been implemented into the finite element software ABAQUS. Table 4 reports the geometrical and material parameters of this case study.

The moment–curvature relationship is depicted in Figure 11 as a moment of 80 N mm has been applied to the beam. Furthermore, Figure 12 shows deflection along the length of SMA beam during loading and unloading. The beam remains deflected about 1.4 mm at the middle part. A good agreement is achieved for analytical results comparing with the finite element method. We also provide stress–height diagram for this case during loading and unloading in Figure 13. Since the material parameters are assumed symmetric and transient zones have not been considered, stress distribution is symmetric as well and three different regions can just appear.

**Example2:** three-point bending test.

In this case study, we have experimentally conducted uniaxial loading–unloading tests (Figure 9a,b) to identify the mechanical properties of Ni-Ti SMA specimens at room temperature (Figure 14). To elucidate the SMA behavior during loading and unloading in variant reorientation zones, we have also conducted two hysteric tests under tensile and compressive forces (Figure 15). According to these experimental tests, the material parameters are presented in Table 5.

After specifying SMA properties, a rectangular beam (10 mm height, 14 mm width) is simply supported at its ends (Figure 9c). Then a three-point bending test with the lower span length of 100 mm and a force of 6.5 kN in the middle section is performed at room temperature. Considering half of the three-point bending problem (Figure 16) and using the equations of Section 5 as well as employing finite difference method (FDM) the load–deflection relationship is illustrated in Figure 17. The maximum difference between current results and experimental data is about 4% in unloading stage. Furthermore, using polynomial approximate functions the relationship between load and deflection for different degrees of the polynomial including 4, 5, and 6 are depicted in Figure 18. It indicates that for degrees of 5 and 6 the results have converged in almost a specific value. However, in a degree of 4 there is about 6% difference compared to the higher degrees, it is still in a good agreement with them.

**Example 3:** four-point bending tests.

According to ASTM E855 − 08, we have finally conducted two four-point bending tests under different applied load for circular beams. The test specimens have a length and diameter of 300 and 20 mm, respectively. The distance between loading and supporting points are 130 and 200 mm (Figure 19). Two LVDTs (linear variable displacement transducer) are installed at the center and one end of the beam to measure the displacement. To assess the bending behavior of the SMA beam under bending before initiation of T- and D-zones favored by tension, applied loads in the first and second tests are, respectively, about 17 and 19 kN.

Using equations in Section 5 and FDM, the load–deflection relationship for the first and second four-point tests has been illustrated in Figure 20 and Figure 21, respectively. There is a good agreement between the results of this work and experimental data. It is worth noting that a maximum 5% difference upon unloading may refer to some simplifications which have been considered in unloading stages.

The relationship between moment and curvature for both tests are shown in Figure 22 and Figure 23. The curvature about 3 m^−1^ may approximately be near to the required stress to initiate D-zone favored by tension ($\kappa_{T2}^{+}$). Making progress in loading stages can increase the slope of moment–curvature relationship. Furthermore, employing Equations (33) and (34), deflection of the SMA beams along the length during loading and unloading stages are determined and indicated in Figure 24 and Figure 25 for first and second four-point bending tests, respectively.

Because of safety, we stopped loading at the curvature about 3.5 m^−1^. To clarify stress–strain behavior of the SMA at this curvature, bending stress during loading and at the end of unloading are obtained and depicted in Figure 26. It shows that most of the fibers in the compressive side are located in D-zone, and some fibers on opposite side are exposed to experience D-zone favored by tension. Upon unloading, comparing the bending stress of present work with general elastic unloading reveals a marked difference between the two theories. It also emphasizes how considering reverse phase transformation during unloading can affect bending behavior of the SME beams. Finally, Figure 27 illustrates the variations of the neutral fiber from the centerline in loading and unloading. It is clear that at the beginning of loading and before reaching stress $\sigma_{P}^{+}$ or $\sigma_{P}^{-}$, stress distribution is totally symmetric. Upon initiation of the detwinning process in tension and then compression sides, the neutral fiber starts moving away from the centroid. Subsequently, more increase in the beam curvature leads to further distance from the centerline. However, the unloading process subtly brings this fiber back to the centroid.

## 9. Summary and Conclusions

This work has analytically and experimentally focused on the flexural behavior of prismatic SMA beams under arbitrary transverse loading. We employ the Euler–Bernouli beam theory and a parametric model to describe asymmetric response of an SMA beam in tension and compression as well as bending behavior. The parametric model provides different elastic properties in martensite twined and detwinned phases. Related equations are initially derived for a prismatic SMA beam under the pure bending problem. To capture general bending equations, we employ pure bending solution and semi-analytical methods. Deflection along the length of the beam can be obtained by using different degrees of polynomial approximation functions. To validate our analytical results, we have experimentally performed three- and four-point bending tests for various SMA beams with different cross-sections at low temperatures. Moment–curvature, stress–height, load–deflection, and variation of the neutral axis from the centerline are reported during loading and unloading. Considering symmetric material behavior, the present model is compared with the available numerical and analytical works, and the result shows an accurate agreement (less than 2% difference) in case of beam curvature and bending moment. Then to validate the effect of asymmetric parameters of our models in terms of predicting bending response of martensitic SMA beams, we initially conduct several three- and four-points bending tests. The obtained analytical results of this work when compared with experimental data in case of load–deflection and moment–curvature relationships show only about 6 and 5%, difference, respectively, proving the high accuracy of the derived governing equations for predicting the SMA beam behavior under general bending. These accurate results stem from assuming the reverse detwinning process during unloading as well as different slope of stress–strain curves when the beam fibers are under loading and unloading conditions. Moreover, this solution could study the distance of neutral fiber from center line during loading and unloading. Since in this paper, the stress–strain of each fiber along the beam height is tracked upon loading and unloading, and the corresponding stress–height diagram to applied bending moment are determined, a deeper practical and physical understanding on the bending problem of SMA beams at low temperatures is achieved. Furthermore, these governing equations can empower the designer to obtain beam deflection and stress distribution of SMA beam with consummate ease.

## Figures and Tables

**Figure 1 materials-14-05415-f001:**
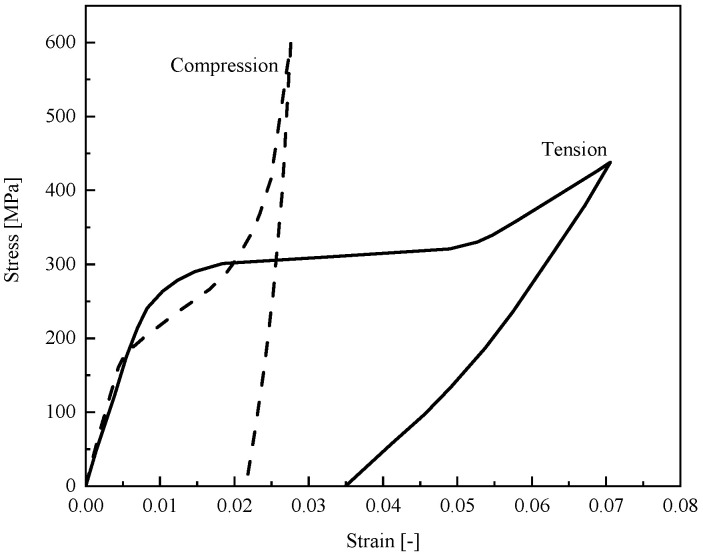
Experimental tension and compression responses for SMA beam at low temperature [36].

**Figure 2 materials-14-05415-f002:**
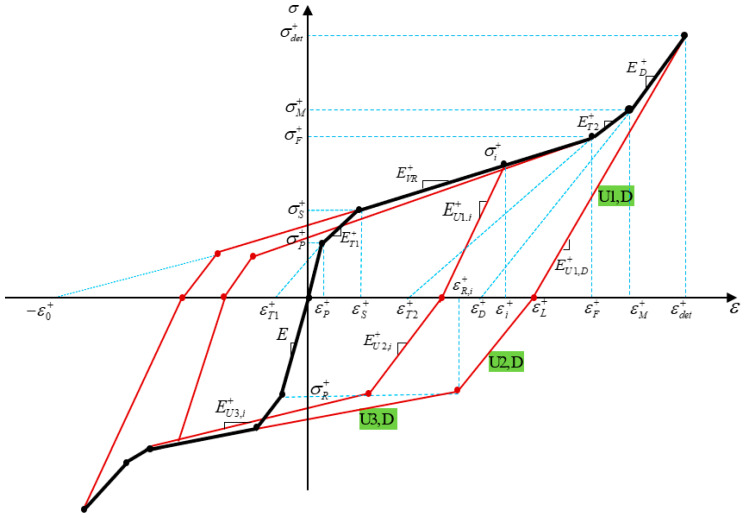
Parametric model to describe the stress–strain relationship of SMA at low temperature.

**Figure 3 materials-14-05415-f003:**
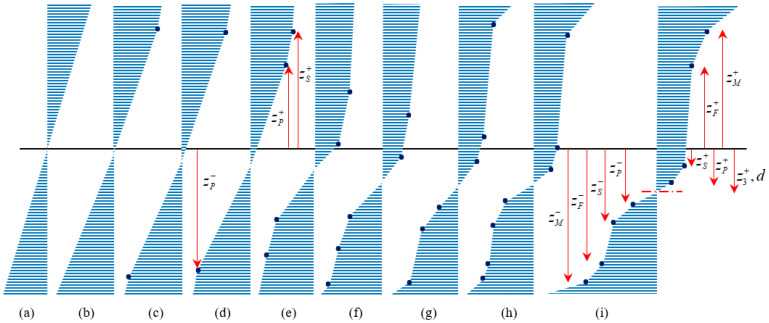
Normal stress–height diagram during different phases of loading, fibers are located in (**a**) elastic zone, (**b**) T1-zone favored by tension, (**c**) T1-zone favored by compression, (**d**) VR-zone favored by tension, (**e**) VR-zone favored by compression, (**f**) T2-zone favored by compression, (**g**) T2-zone favored by tension, (**h**) D-zone favored by compression, (**i**) D-zone favored by tension.

**Figure 4 materials-14-05415-f004:**
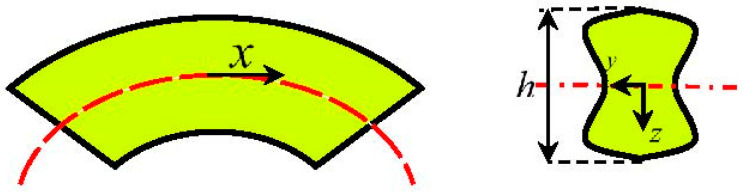
Schematic of the SMA beam and its cross-section under bending.

**Figure 5 materials-14-05415-f005:**
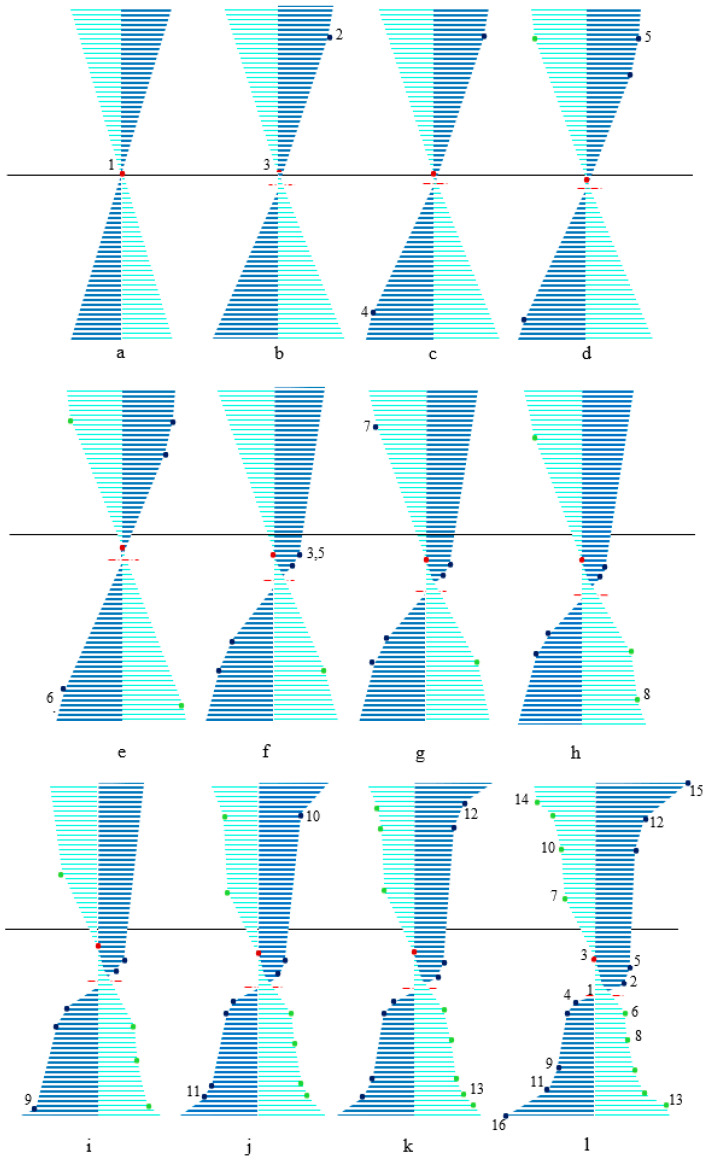
Schematic of stress–height variations for an SMA beam at low temperature during unloading in (**a**) elastic zone, (**b**) T1-zone favored by tension, (**c**) T1-zone favored by compression, (**d**) VR-zone favored by tension, (**e**) VR-zone favored by compression, (**f**) when $z_{S}^{+}$ is coincident with $d_{u}$, (**g**) due to reverse T-zone, (**h**) due to reverse VR-zone, (**i**) T2-zone favored by compression, (**j**) T2-zone favored by tension, (**k**) D-zone favored by compression, (**l**) D-zone favored by tension.

**Figure 6 materials-14-05415-f006:**
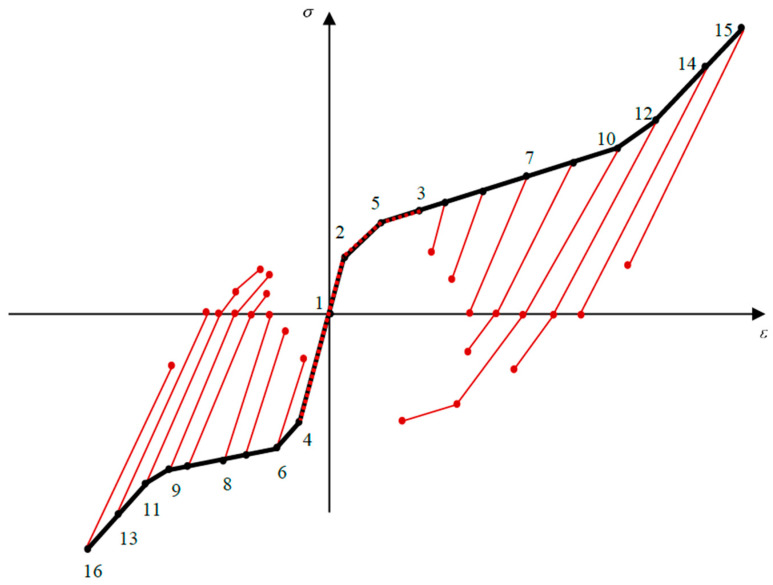
Stress–strain curve for SMA beam at low temperature upon unloading.

**Figure 7 materials-14-05415-f007:**
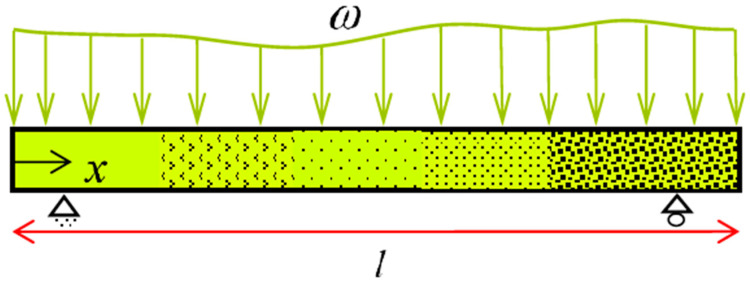
A prismatic SMA beam subjected to transverse loading (general bending).

**Figure 8 materials-14-05415-f008:**
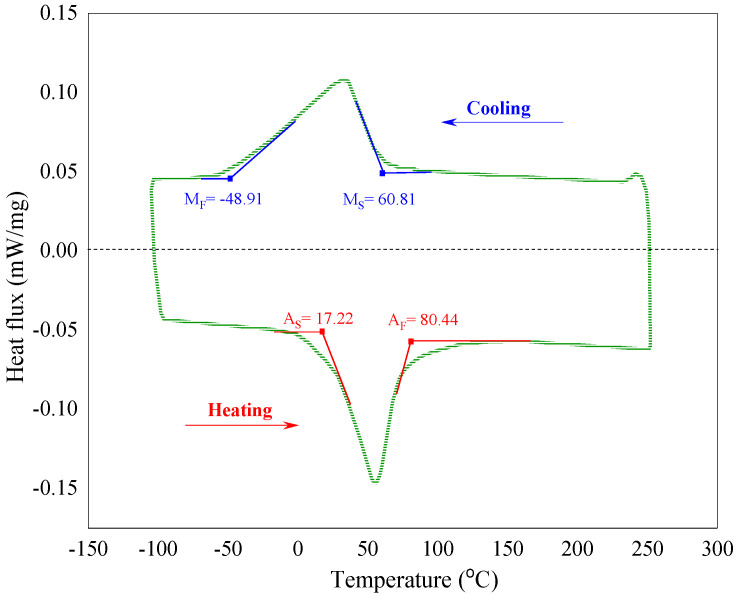
DSC curves of the martensite SMA beam.

**Figure 9 materials-14-05415-f009:**
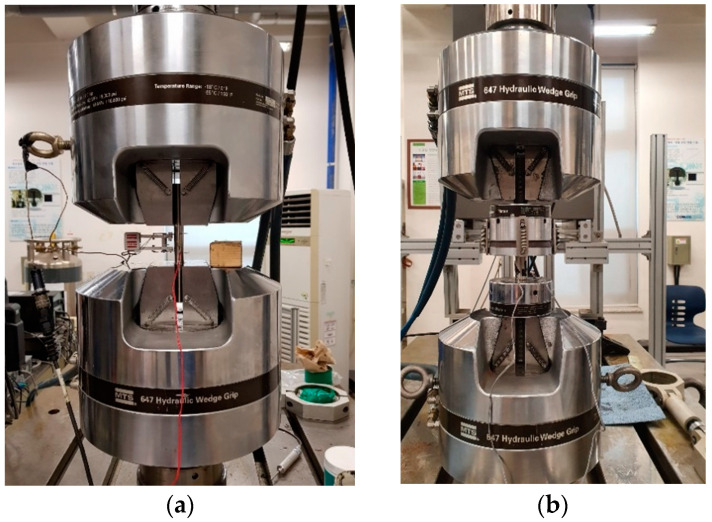
Experimental tests setup. (**a**) Tensile test, (**b**) compression test, (**c**) bending test for rectangular beam, (**d**) bending test for circular beam, (**e**) circular beams before bending tests, (**f**) after bending tests.

**Figure 10 materials-14-05415-f010:**
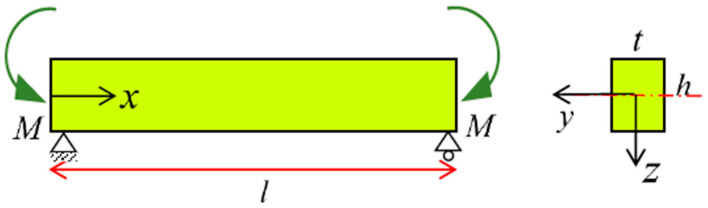
Rectangular beam subjected to the pure bending.

**Figure 11 materials-14-05415-f011:**
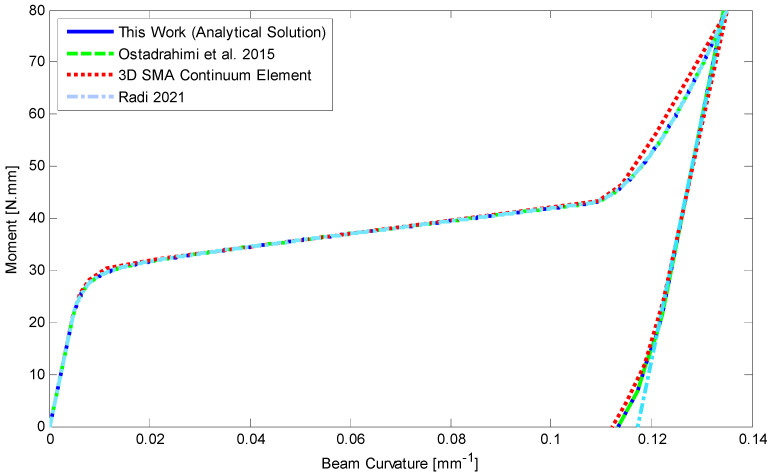
Comparison of analytical results considering symmetric material parameters and finite element for a rectangular SMA beam subjected to a moment of 80 N.mm at −40 °C, [28,35].

**Figure 12 materials-14-05415-f012:**
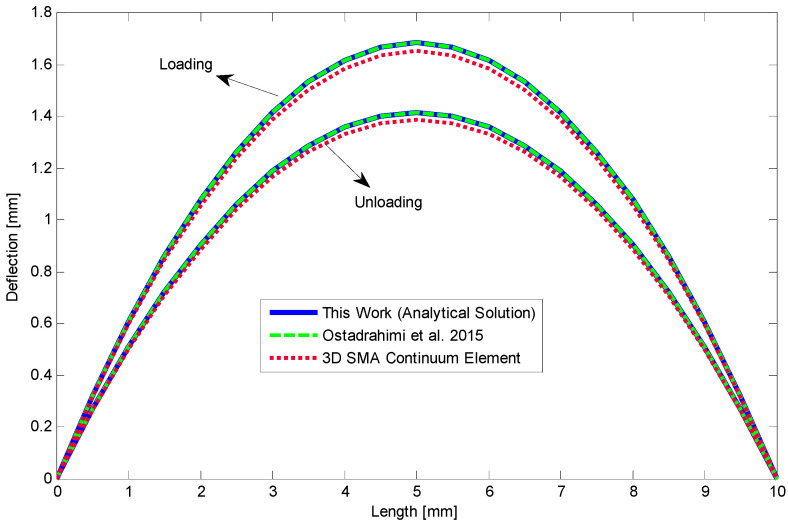
Deflection versus length of the SMA beam with symmetric material properties in loading and unloading [28].

**Figure 13 materials-14-05415-f013:**
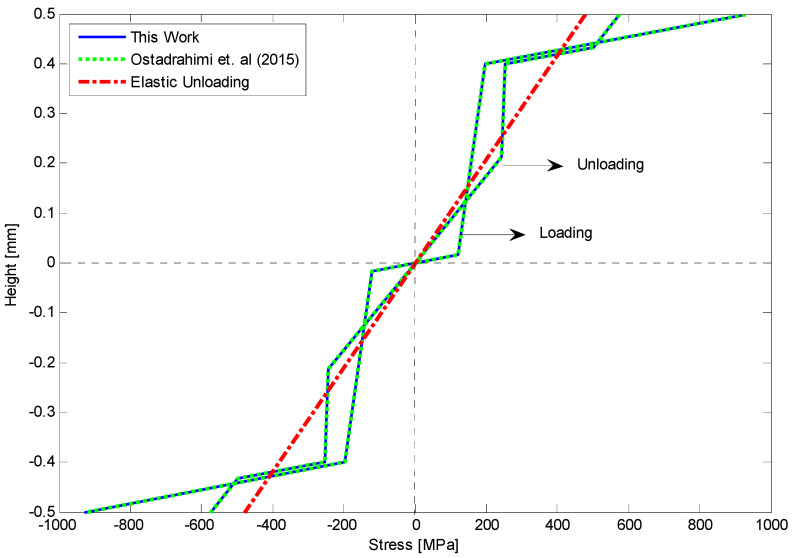
Comparison of stress–height relationship upon loading and unloading with elastic unloading for pure bending problem with symmetric tension–compression response [28].

**Figure 14 materials-14-05415-f014:**
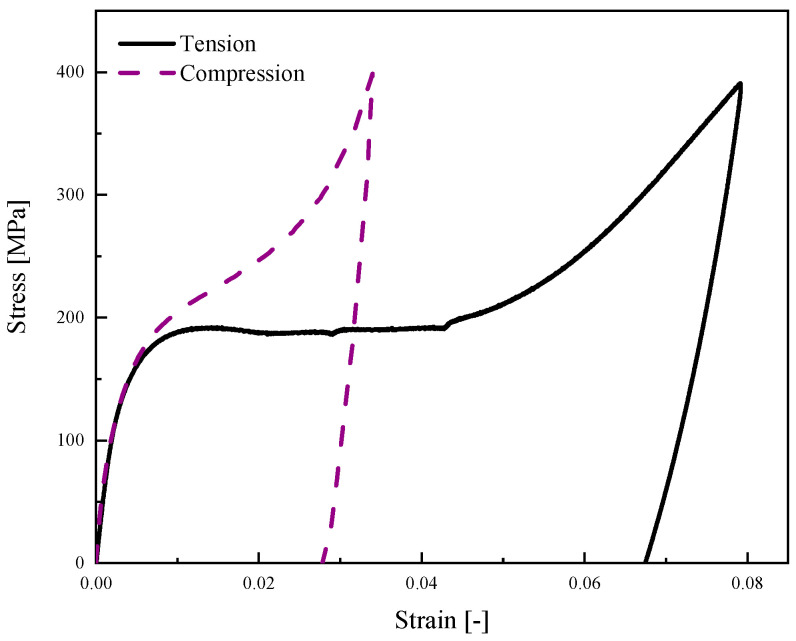
Monotonic tensile and compression tests for martensite SMA beam at room temperature.

**Figure 15 materials-14-05415-f015:**
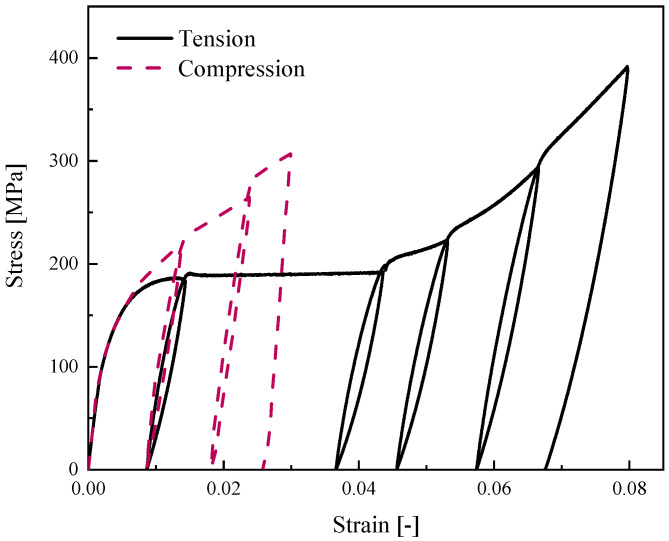
Hysteric tensile and compression tests for martensite SMA beam at room temperature.

**Figure 16 materials-14-05415-f016:**
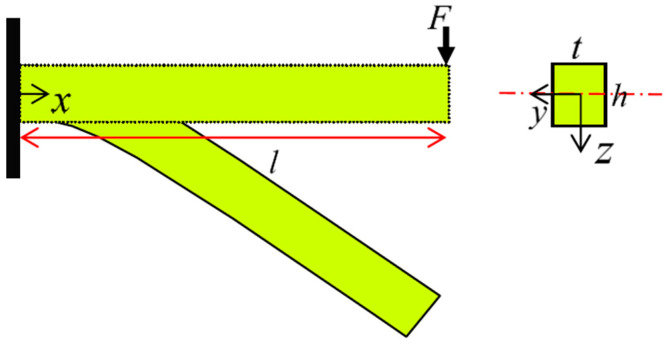
The cantilever (half of the three-point bending model) beam subjected to transverse loading at free end.

**Figure 17 materials-14-05415-f017:**
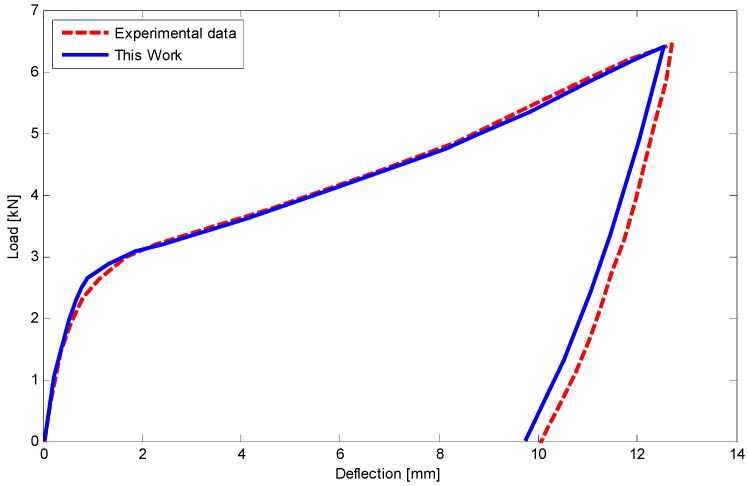
Comparison of load and deflection for a rectangular SMA beam subjected to the three-point bending test at room temperature.

**Figure 18 materials-14-05415-f018:**
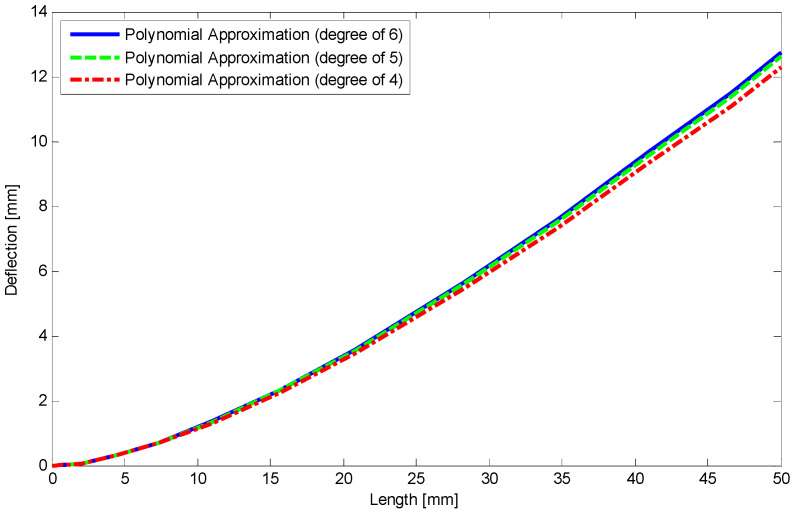
Comparison of different degrees of polynomial approximation function for deflection of SMA beam along the length.

**Figure 19 materials-14-05415-f019:**
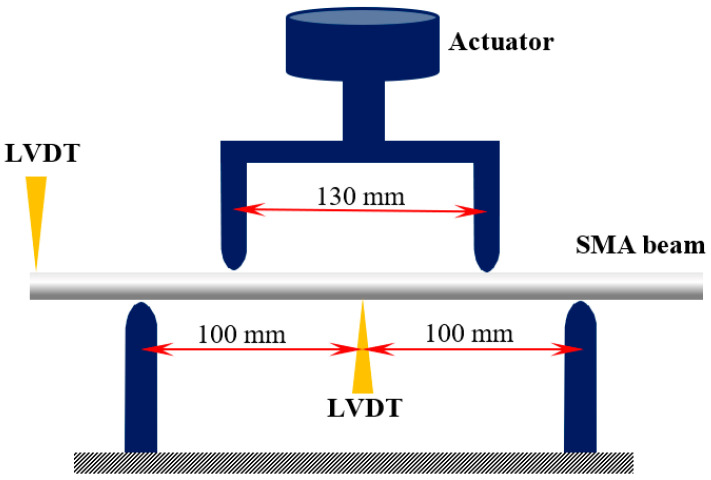
Dimension and schematically experimental setup for four-point bending tests.

**Figure 20 materials-14-05415-f020:**
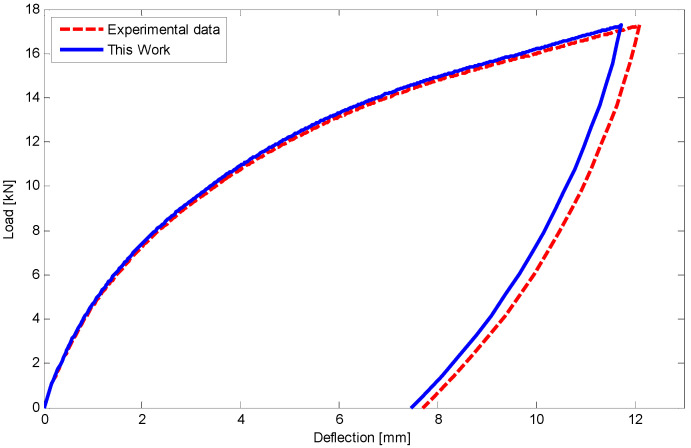
Comparison of load versus deflection for a circular SMA beam under four-point bending test with force of 17 kN at room temperature.

**Figure 21 materials-14-05415-f021:**
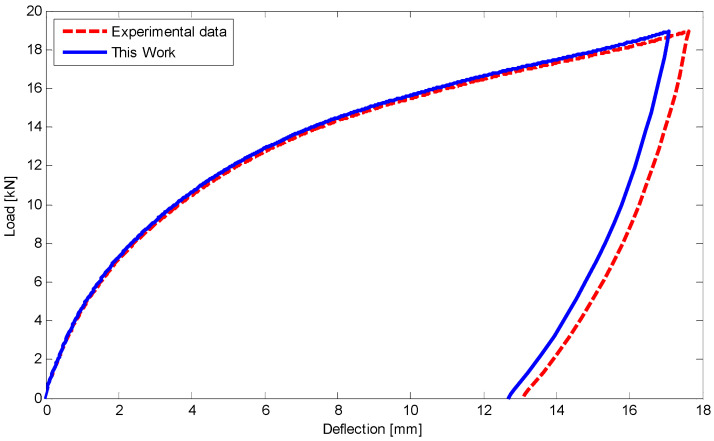
Comparison of load versus deflection for a circular SMA beam subjected to four-point bending test with force about 19 kN at room temperature.

**Figure 22 materials-14-05415-f022:**
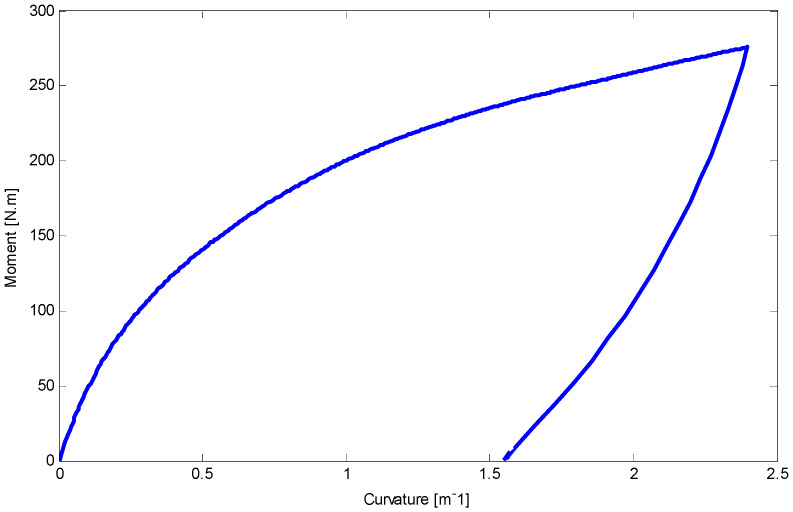
The relationship between curvature and moment for the circular SMA beam under four-point bending test with the force of 17 kN at room temperature.

**Figure 23 materials-14-05415-f023:**
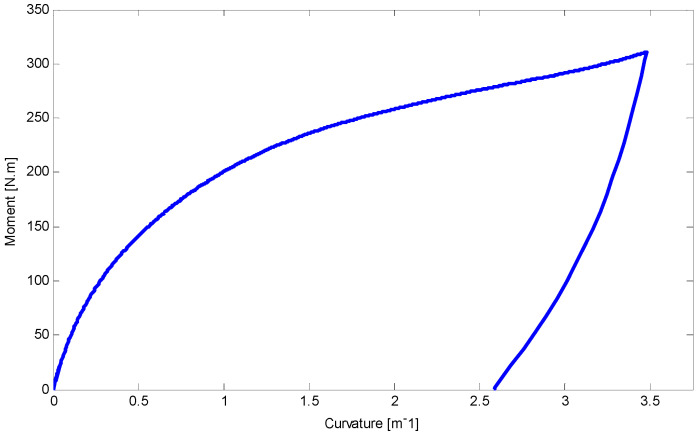
Curvature and moment relationship for the circular SMA beam subjected to four-point bending test with the force about 19 kN at room temperature.

**Figure 24 materials-14-05415-f024:**
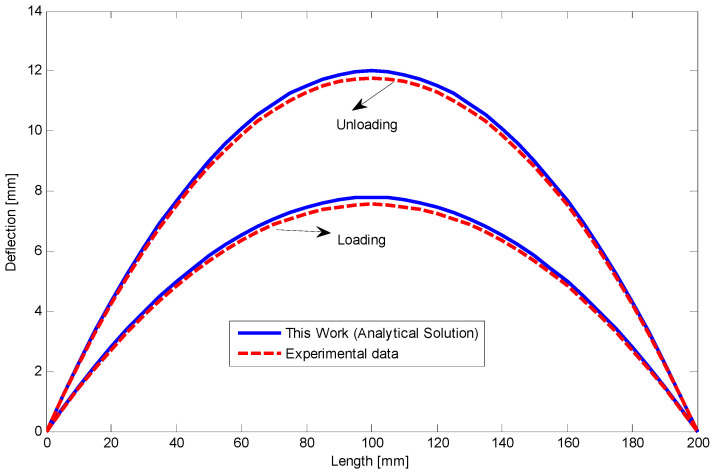
The relationship between deflection and length at the end of loading and unloading for the circular SMA beam under four-point bending test with the force of 17 kN.

**Figure 25 materials-14-05415-f025:**
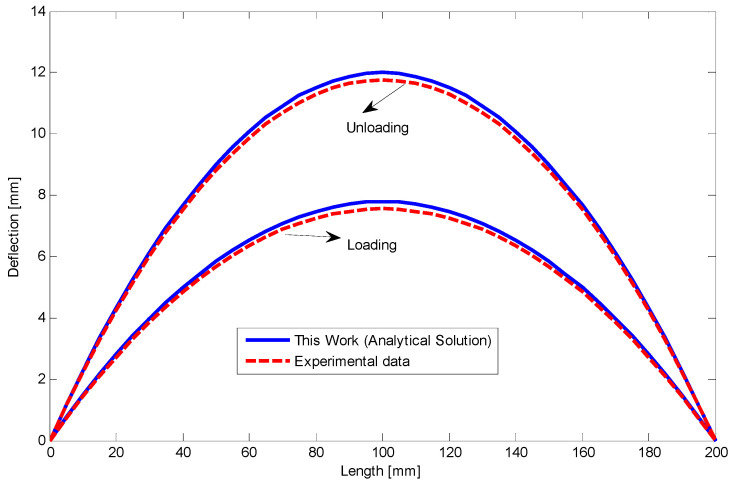
The relationship between deflection and length at the end of loading and unloading for the circular SMA beam under four-point bending test with the force about 19 kN.

**Figure 26 materials-14-05415-f026:**
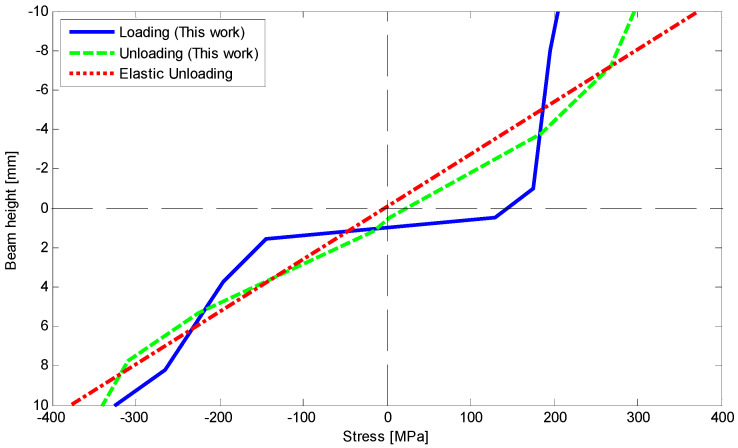
Comparison of stress–height relationship upon loading and unloading with elastic unloading for four-point bending test.

**Figure 27 materials-14-05415-f027:**
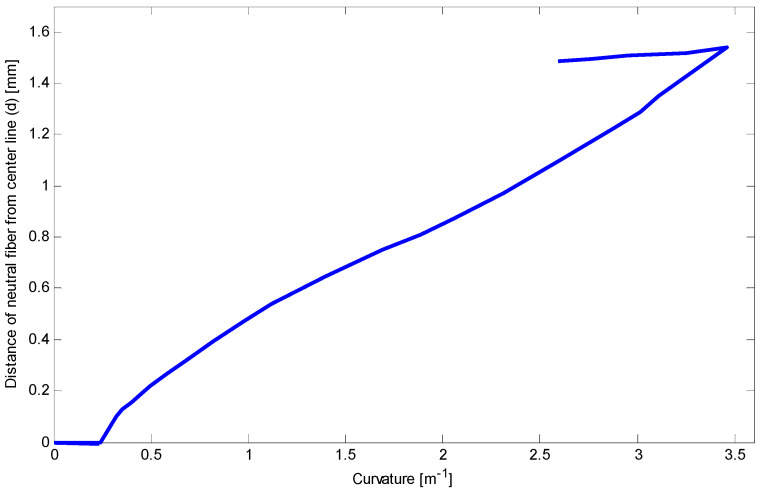
Relationship between curvature and distance of neutral fiber from the centerline during loading and unloading.

**Table 1 materials-14-05415-t001:** Material parameters and constitutive equations related to each zone in tension and compression during loading.

Parameters	Zones			
	Elastic	Tj-Zone	VR-Zone	D-Zone
**Transformation strain**	*0*	$\varepsilon_{Tj}^{\pm }$	$\varepsilon_{VR}^{\pm }$	$\varepsilon_{D}^{\pm }$
**Elastic modulus**	*E*	$E_{Tj}^{\pm }$	$E_{VR}^{\pm }$	$E_{D}^{\pm }$
**Constitutive equation**	σ=Eε	$\sigma =E_{Tj}^{\pm }(\varepsilon -\varepsilon_{Tj}^{\pm })$	$\sigma =E_{VR}^{\pm }(\varepsilon -\varepsilon_{VR}^{\pm })$	$\sigma =E_{D}^{\pm }(\varepsilon -\varepsilon_{D}^{\pm })$

**Table 2 materials-14-05415-t002:** Elastic modulus and transformation strain related to each zone.

Zones	Elastic Modulus	Transformation Strain	Parameters
Elastic	$E=\frac{\sigma_{P}^{+}}{\varepsilon_{P}^{+}}$	0	$\sigma_{P}^{\pm }$($\varepsilon_{P}^{\pm }$): proportional limit stress (strain)
T1-zone	$E_{T1}^{\pm }=\frac{\sigma_{S}^{\pm }-\sigma_{P}^{\pm }}{\varepsilon_{S}^{\pm }-\varepsilon_{P}^{\pm }}$	$\varepsilon_{T1}^{\pm }=\varepsilon_{P}^{\pm }-\frac{\sigma_{P}^{\pm }}{E_{T1}^{\pm }}$	$\sigma_{S}^{\pm }$ ($\varepsilon_{S}^{\pm }$): VR-zone start stress (strain)
VR-zone	$E_{VR}^{\pm }=\frac{\sigma_{F}^{\pm }-\sigma_{S}^{\pm }}{\varepsilon_{F}^{\pm }-\varepsilon_{S}^{\pm }}$	$\varepsilon_{VR}^{\pm }=\varepsilon_{S}^{\pm }-\frac{\sigma_{S}^{\pm }}{E_{VR}^{\pm }}$	$\sigma_{F}^{\pm }$ ($\varepsilon_{F}^{\pm }$): VR-zone finish stress (strain)
T2-zone	$E_{T2}^{\pm }=\frac{\sigma_{M}^{\pm }-\sigma_{F}^{\pm }}{\varepsilon_{M}^{\pm }-\varepsilon_{F}^{\pm }}$	$\varepsilon_{T2}^{\pm }=\varepsilon_{F}^{\pm }-\frac{\sigma_{F}^{\pm }}{E_{T2}^{\pm }}$	$\sigma_{M}^{\pm }$($\varepsilon_{M}^{\pm }$): maximum detwinning stress (strain)
D-zone	$E_{D}^{\pm }=\frac{\sigma_{det}^{\pm }-\sigma_{M}^{\pm }}{\varepsilon_{det}^{\pm }-\varepsilon_{M}^{\pm }}$	$\varepsilon_{D}^{\pm }=\varepsilon_{M}^{\pm }-\frac{\sigma_{M}^{\pm }}{E_{D}^{\pm }}$	$\sigma_{det}^{\pm }$($\varepsilon_{det}^{\pm }$): stress (strain) in D-zone

**Table 3 materials-14-05415-t003:** General moment and force equations for the rectangular beam subjected to the pure bending problem.

Parameters	Values
Loading moment	$\begin{array}{l} \frac{1}{3}\sigma_{P}^{+}{\left(d-z_{P}^{+}\right)}^{2}+\sigma_{P}^{+}(z_{P}^{+}-z_{s}^{+})\left(-\frac{1}{2}z_{P}^{+}-\frac{1}{2}z_{S}^{+}+d\right)\\ +\frac{1}{2}(\sigma_{S}^{+}-\sigma_{P}^{+})\;(z_{P}^{+}-z_{S}^{+})\left(-\frac{1}{3}z_{P}^{+}-\frac{2}{3}z_{S}^{+}+d\right)+\\ \sigma_{S}^{+}(z_{S}^{+}-z_{F}^{+})\left(-\frac{1}{2}z_{S}^{+}-\frac{1}{2}z_{F}^{+}+d\right)+\frac{1}{2}(\sigma_{F}^{+}-\sigma_{S}^{+})\;(z_{S}^{+}-z_{F}^{+})\left(-\frac{1}{3}z_{S}^{+}-\frac{2}{3}z_{F}^{+}+d\right)\\ +\sigma_{F}^{+}(z_{F}^{+}-z_{M}^{+})\left(-\frac{1}{2}z_{F}^{+}-\frac{1}{2}z_{M}^{+}+d\right)+\sigma_{M}^{+}(z_{M}^{+}+\frac{h}{2})\left(-\frac{1}{2}z_{M}^{+}+\frac{h}{4}+d\right)\\ +\frac{1}{2}(\sigma_{M}^{+}-\sigma_{F}^{+})\;(z_{F}^{+}-z_{M}^{+})\left(-\frac{1}{3}z_{F}^{+}-\frac{2}{3}z_{M}^{+}+d\right)+\frac{1}{2}(\sigma_{i}^{+}-\sigma_{M}^{+})\;(z_{M}^{+}+\frac{h}{2})\left(-\frac{1}{3}z_{M}^{+}+\frac{h}{3}+d\right)\\ +\frac{1}{3}\sigma_{P}^{-}{\left(z_{P}^{-}-d\right)}^{2}+\sigma_{P}^{-}(z_{S}^{-}-z_{P}^{-})\left(\frac{1}{2}z_{S}^{-}+\frac{1}{2}z_{P}^{-}-d\right)+\frac{1}{2}(\sigma_{S}^{-}-\sigma_{P}^{-})\;(z_{S}^{-}-z_{P}^{+})\left(\frac{2}{3}z_{S}^{-}+\frac{1}{3}z_{P}^{-}-d\right)\\ +\sigma_{S}^{-}(z_{F}^{-}-z_{S}^{-})\left(\frac{1}{2}z_{F}^{-}+\frac{1}{2}z_{S}^{-}-d\right)+\frac{1}{2}(\sigma_{F}^{-}-\sigma_{S}^{-})\;(z_{F}^{-}-z_{S}^{-})\left(\frac{2}{3}z_{F}^{-}+\frac{1}{3}z_{S}^{-}-d\right)+\\ +\sigma_{F}^{-}(z_{M}^{-}-z_{F}^{-})\left(\frac{1}{2}z_{M}^{-}+\frac{1}{2}z_{F}^{-}-d\right)+\sigma_{M}^{-}(\frac{h}{2}-z_{M}^{-})\left(\frac{h}{4}+\frac{1}{2}z_{M}^{-}-d\right)\\ \frac{1}{2}(\sigma_{M}^{-}-\sigma_{F}^{-})\;(z_{M}^{-}-z_{F}^{-})\left(\frac{2}{3}z_{M}^{-}+\frac{1}{3}z_{F}^{-}-d\right)+\frac{1}{2}(\sigma_{i}^{-}-\sigma_{M}^{-})\;(\frac{h}{2}-z_{M}^{-})\left(\frac{h}{3}+\frac{1}{3}z_{M}^{-}-d\right) \end{array}$
Force equilibrium in loading	$\begin{array}{l} \sigma_{P}^{+}(d-z_{P}^{+})-(\sigma_{F}^{-}+\sigma_{S}^{-})\;(z_{F}^{-}-z_{S}^{-})+(\sigma_{S}^{+}+\sigma_{P}^{+})\;(z_{P}^{+}-z_{S}^{+})+(\sigma_{F}^{+}+\sigma_{S}^{+})\;(z_{S}^{+}-z_{F}^{+})\\ -\sigma_{P}^{-}(z_{P}^{-}-d)+(\sigma_{M}^{+}+\sigma_{F}^{+})\;(z_{F}^{+}-z_{M}^{+})-(\sigma_{M}^{-}+\sigma_{F}^{-})\;(z_{M}^{-}-z_{F}^{-})\\ +(\sigma_{i}^{+}+\sigma_{M}^{+})\;(z_{M}^{+}+\frac{h}{2})-(\sigma_{S}^{-}+\sigma_{P}^{-})\;(z_{S}^{-}-z_{P}^{-})-(\sigma_{i}^{-}+\sigma_{M}^{-})\;(\frac{h}{2}-z_{M}^{-}) \end{array}$
Unloading moment	$\begin{array}{c} \frac{1}{3}\Delta \sigma_{2}{\left(\frac{\sigma_{P}^{+}}{\kappa E}+d-du\right)}^{2}\\ \Delta \sigma_{2}\left(-\frac{\sigma_{S}^{-}}{\kappa E_{T1}^{+}}+\frac{\sigma_{P}^{+}}{\kappa E}-\frac{\varepsilon_{T1}^{-}}{\kappa }\right)\left(-\frac{1}{2}\frac{\sigma_{S}^{-}}{\kappa E_{T1}^{+}}-\frac{1}{2}\frac{\varepsilon_{T1}^{-}}{\kappa }-\frac{1}{2}\frac{\sigma_{P}^{-}}{\kappa E}+d-d_{u}\right)\\ +\frac{1}{2}\;(\Delta \sigma_{6}-\Delta \sigma_{2})\;\left(-\frac{\sigma_{S}^{-}}{\kappa E_{T1}^{+}}+\frac{\sigma_{P}^{+}}{\kappa E}-\frac{\varepsilon_{T1}^{-}}{\kappa }\right)\left(-\frac{2}{3}\frac{\sigma_{S}^{-}}{\kappa E_{T1}^{+}}-\frac{2}{3}\frac{\varepsilon_{T1}^{-}}{\kappa }-\frac{1}{3}\frac{\sigma_{P}^{+}}{\kappa E}+d-d_{u}\right)\\ +\Delta \sigma_{6}\left(z_{8}-d+\frac{\sigma_{S}^{-}}{\kappa E_{T1}^{-}}+\frac{\sigma_{P}^{+}}{\kappa E}-\frac{\varepsilon_{T1}^{-}}{\kappa }\right)\left(\frac{z_{8}+d}{2}-\frac{1}{2}\frac{\sigma_{S}^{-}}{\kappa E_{T1}^{-}}-\frac{1}{2}\frac{\varepsilon_{T1}^{-}}{\kappa }-d_{u}\right)\\ +\frac{1}{2}\;(\Delta \sigma_{8}-\Delta \sigma_{6})\left(z_{8}-d+\frac{\sigma_{S}^{-}}{\kappa E_{T1}^{-}}+\frac{\varepsilon_{T1}^{-}}{\kappa }\right)\;\left(\frac{2}{3}z_{8}+\frac{1}{3}d-\frac{1}{3}\frac{\sigma_{S}^{-}}{\kappa E_{T1}^{-}}-\frac{1}{3}\frac{\varepsilon_{T1}^{-}}{\kappa }-d_{u}\right)\\ +\frac{1}{3}\;(\Delta \sigma_{9}-\Delta \sigma_{8})\left(d-\frac{\sigma_{S}^{-}}{\kappa E_{T1}^{-}}-\frac{\varepsilon_{S}^{-}-\varepsilon_{F}^{-}+\varepsilon_{T1}^{-}}{\kappa }-z_{8}\right)\\ \left(d-\frac{\sigma_{S}^{-}}{\kappa E_{T1}^{-}}-\frac{\varepsilon_{S}^{-}-\varepsilon_{F}^{-}+\varepsilon_{T1}^{-}}{\kappa }+\frac{1}{2}z_{8}-\frac{3}{2}d_{u}\right)\\ +\Delta \sigma_{8}\left(d-\frac{\sigma_{S}^{-}}{\kappa E_{T1}^{-}}-\frac{\varepsilon_{S}^{-}-\varepsilon_{F}^{-}+\varepsilon_{T1}^{-}}{\kappa }-z_{8}\right)\left(\frac{z_{8}+d}{2}-\frac{1}{2}\frac{\sigma_{S}^{-}}{\kappa E_{T1}^{-}}-\frac{1}{2}\frac{\varepsilon_{T1}^{-}-\varepsilon_{F}^{-}+\varepsilon_{T1}^{-}}{\kappa }-d_{u}\right)\\ +\frac{1}{3}\;(\Delta \sigma_{11}-\Delta \sigma_{9})\left(\frac{h}{2}-z_{P}^{-}-z_{S}^{-}-z_{F}^{-}-d+\frac{\sigma_{S}^{-}}{\kappa E_{T1}^{-}}-\frac{\varepsilon_{S}^{-}-\varepsilon_{F}^{-}+\varepsilon_{T1}^{-}}{\kappa }\right)\\ \left(\frac{h+d}{2}-z_{P}^{-}-z_{S}^{-}-z_{F}^{-}-\frac{\varepsilon_{S}^{-}-\varepsilon_{F}^{-}+\varepsilon_{T1}^{-}}{2\kappa }-\frac{\sigma_{S}^{-}}{2\kappa E_{T1}^{-}}-\frac{3}{2}d_{u}\right)\\ +\Delta \sigma_{9}\left(\frac{h}{2}-z_{P}^{-}-z_{S}^{-}-z_{F}^{-}-d+\frac{\sigma_{S}^{-}}{\kappa E_{T1}^{-}}+\frac{\varepsilon_{S}^{-}-\varepsilon_{F}^{-}+\varepsilon_{T1}^{-}}{\kappa }\right)\\ \left(\frac{d-z_{P}^{-}-z_{S}^{-}-z_{F}^{-}}{2}+\frac{h}{4}-\frac{1}{2}\frac{\sigma_{S}^{-}}{\kappa E_{T1}^{-}}-\frac{1}{2}\frac{\varepsilon_{S}^{-}-\varepsilon_{F}^{-}+\varepsilon_{T1}^{-}}{\kappa }-d_{u}\right)\\ +\frac{1}{3}\;(\Delta \sigma_{13}-\Delta \sigma_{11})\left(z_{13}-\frac{h}{2}+z_{P}^{-}+z_{S}^{-}+z_{F}^{-}\right)\left(z_{13}+\frac{h}{4}-2(z_{P}^{-}+z_{S}^{-}+z_{F}^{-})\right)\\ +\Delta \sigma_{11}\left(z_{13}-\frac{h}{2}+z_{P}^{-}+z_{S}^{-}+z_{F}^{-}\right)\left(z_{13}+\frac{h}{4}-\frac{z_{P}^{-}+z_{S}^{-}+z_{F}^{-}}{2}\right)\\ +\frac{1}{3}\;(\Delta \sigma_{h/2}-\Delta \sigma_{13}){\left(\frac{h}{2}-z_{13}\right)}^{}\left(\frac{h+z_{13}}{3}\right)+\frac{1}{2}\Delta \sigma_{h/2}{\left(\frac{h}{2}-z_{13}\right)}^{2} \end{array}$
Force equilibrium in unloading	$\begin{array}{l} \Delta \sigma_{7}(du-z_{7})+(\Delta \sigma_{10}+\Delta \sigma_{7})\;\left(z_{7}-d+\frac{\sigma_{S}^{+}}{\kappa E_{T1}^{+}}+\frac{\varepsilon_{S}^{+}-\varepsilon_{F}^{+}-\varepsilon_{T1}^{+}}{\kappa }\right)\\ +(\Delta \sigma_{12}+\Delta \sigma_{10})\left(d-\frac{\sigma_{S}^{+}}{\kappa E_{T1}^{+}}-\frac{\varepsilon_{S}^{+}+\varepsilon_{T1}^{+}-\varepsilon_{F}^{+}}{\kappa }-\frac{h}{2}+z_{P}^{+}+z_{S}^{+}+z_{F}^{+}\right)\\ -\Delta \sigma_{2}\left(d-d_{u}-\frac{\sigma_{P}^{+}}{\kappa E_{}^{+}}\right)-(\Delta \sigma_{2}+\Delta \sigma_{6})\left(\frac{\sigma_{P}^{+}}{\kappa E_{}^{+}}-\frac{\sigma_{S}^{-}}{\kappa E_{T1}^{-}}-\frac{\varepsilon_{T1}^{-}}{\kappa }\right)\\ +(\Delta \sigma_{14}+\Delta \sigma_{12})\left(\frac{h}{2}-(z_{P}^{+}+z_{S}^{+}+z_{F}^{+}+z_{14})\right)+(\Delta \sigma_{12}+\Delta \sigma_{-h/2})\;\left(\frac{h}{2}+z_{14}\right)\\ -(\Delta \sigma_{9}+\Delta \sigma_{11})\left(\frac{h}{2}-(z_{P}^{-}+z_{S}^{-}+z_{F}^{-}+d)+\frac{\sigma_{S}^{-}}{\kappa E_{T1}^{-}}+\frac{\varepsilon_{S}^{-}-\varepsilon_{F}^{-}+\varepsilon_{T1}^{-}}{\kappa }\right)\\ +(\Delta \sigma_{8}+\Delta \sigma_{6})\;\left(z_{8}-d+\frac{\sigma_{S}^{-}}{\kappa E_{T1}^{-}}+\frac{\varepsilon_{T1}^{+}}{\kappa }\right)-(\Delta \sigma_{h/2}+\Delta \sigma_{13})\;\left(\frac{h}{2}-z_{13}\right)\\ -(\Delta \sigma_{8}+\Delta \sigma_{9})\;\left(d-z_{8}-\frac{\sigma_{S}^{-}}{\kappa E_{T1}^{-}}-\frac{\varepsilon_{S}^{+}-\varepsilon_{F}^{+}+\varepsilon_{T1}^{+}}{\kappa }\right)-(\Delta \sigma_{11}+\Delta \sigma_{13})\;\left(z_{13}-\frac{h}{2}+z_{P}^{-}+z_{S}^{-}+z_{F}^{-}\right) \end{array}$

**Table 4 materials-14-05415-t004:** Model parameters of the SMA beam.

**Material Parameter**	**Value**	**Unit**
$\sigma_{P}^{},\sigma_{S}^{}$	122.5	MPa
$\sigma_{F}^{},\sigma_{M}^{}$	197.5	MPa
$\varepsilon_{P}^{},\varepsilon_{S}^{}$	0.23%	-
$\varepsilon_{L}^{}$	5%	-
$\varepsilon_{F}^{},\varepsilon_{M}^{}$	5.37%	-
**Geometric Parameter**	**Value**	**Unit**
*h*	1	mm
*t*	1	mm
*l*	10	mm

**Table 5 materials-14-05415-t005:** Material properties extracted from experimental tests.

$\sigma_{P}^{\pm }\;(\mathrm{MPa})$	$\sigma_{S}^{\pm }\;(\mathrm{MPa})$	$\sigma_{F}^{\pm }\;(\mathrm{MPa})$	$\sigma_{M}^{\pm }\;(\mathrm{MPa})$	$\varepsilon_{P}^{\pm }\;(\%)$	$\varepsilon_{S}^{\pm }\;(\%)$	$\varepsilon_{F}^{\pm }\;(\%)$	$\varepsilon_{M}^{\pm }\;(\%)$	$\varepsilon_{L}^{}\;(\%)$
+130 −144	+180 −195.6	+197 −265.6	+247 −329.3	+0.29 −0.32	+0.7 −0.76	+4.3 −2.3	+6 −3	+6.9 −2.4

## Data Availability

Not applicable.

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
