# Peer review of "Effect of Tension-Compression Asymmetry Response on the Bending of Prismatic Martensitic SMA Beams: Analytical and Experimental Study"

_materials, 2021, doi:10.3390/ma14185415_

Round 1

Reviewer 1 Report

The submitted manuscript entitled “Effect of Tension-Compression Asymmetry Response on the Bending of Prismatic Martensitic SMA Beams: Analytical and Experimental Study” has some interesting results. The Authors of this manuscript developed an analytical model of bending as well as a semi-analytical prediction of the deflection of the shape memory alloy prismatic beams in the martensite phase. Additionally, experimental studies were carried out to verify the analytical bending model. To validate the current work, the results of the pure bending problem were initially compared with the numerical model using the finite element method, which assumed symmetrical behaviour of the material.

Attention:

  1. The authors on page 17 in line 381 wrote “… we initially compare the results of the pure bending problem with the finite element method in which symmetric material behavior is assumed”. The Finite Element Method (FEM) is a numerical method for solving differential equations. I understand that the authors were talking about a numerical model or commercial software (such as ABAQUS or ANSYS), and FEM is a mental shortcut. However, I would recommend that you use a specific software name at work.

Author Response

The response file is attached.

Reviewer 2 Report

Dear Authors,

Congratulations to the authors on a truly comprehensive analytical and numerical study of the behavior of martensitic SMA supported by experimental results. The manuscript is excellently conceived and in accordance with the results and the presented conclusions can really contribute to a good prediction of the flexural behavior of prismatic SMA beams under arbitrary transverse loading.

In order to improve the quality of the article, the authors are asked to consider the following comments:

Page 5, Line 147:  The explanation of the parameter eL in equation 3 is missing.

Page 6, Line 163:  Please correct the illegible markings in Figure 3.

Pages 10 and 11,  Lines from 259 to 262: Why are Figs. 5a-l not arranged correctly?

Page 12, Line 277:  The markings in Fig.6 are illegible. Please correct.

Pages 15, 16, 17: In Chapter 6. Experimental tests, it is necessary to clearly indicate on which material the  tests were carried out, with showing its chemical composition.

Data on devices (manufacturer, type, country) on which differential scanning calorimetry, tensile and compressive tests and bending test were carried out are missing.

Exactly how many samples of circular / rectangular cross-section were subjected to three- and four-points bending tests?

In this part it should be stated how (with which devices) the deflection during bending was measured.

Figure 9e is not required and can be removed. Figure marks need to be aligned.

Pages 21 and 22,  The solid purple line denoting "This work (Analytical Solution)" in legend of Figures 11, 12, 13 should be replaced with a dashed curve.

Page 23, Line 433:  Please, separate the measurement unit  from the value.

Page 24, Line 456:  The full name for the abbreviation LVDT is missing. Please correct.

Page 25, Line 460:  The markings in Fig.19 are illegible. Please correct.

Page 28, Line 508:  The measurement unit of curvature  must be specified in the same way (1/m → m-1).

How is it possible that when carried out experimental tests (tension and compression loading–unloading test, three-point bending test, four-point bending test) at room temperature (Figs. 14, 15, 17, 20, 22, 23, 24, 25) the material have the martensitic structure with respect to the transformation temperatures which are shown in Figure 8?

Please align the labels of the all equations with respect to the right margin.

References

[5] and [24]: "niti" should be replaced with "NiTi"

[6], [9], [13], [38]: Correct first and last page citations (from - to).

[15]: Correct the typo (stretchingbending?)

[42: Delete the year of publication (stated twice)

Reviewer

Author Response

The response file is attached.

Reviewer 3 Report

The article is devoted to the work on the analytical derivation of the bending equations, as well as the prediction of the deflection of prismatic SMA beams in the martensitic phase. The authors' model takes into account various parameters of the material in the martensitic soldered and split twin phases, as well as the elastic modulus depending on the course of the twinning process. In addition, the model takes into account different loading and unloading slopes in the separated martensite phases, which are facilitated by tension and compression.

The practical significance of the work is obvious. Shape memory alloy (SMA) is one of the smart materials with two well-known characteristics and a highly non-linear stress-strain curve. The manifestation of pseudo-elasticity (PE) and shape memory effect (SME), as well as thermomechanically coupled reactions, have widely opened up wide opportunities for the use of SMA in various industrial sectors, including biomedical, aerospace and civil structures. However, the article would benefit even more from a more detailed description of the application of such materials in practice. We wish the authors to give detailed examples of their application in practice.

The scientific significance of the article is high. The authors demonstrated excellent skills in mathematical methods and knowledge of materials science fundamentals

In addition, the article contains minor flaws that need to be finalized and corrected.

Author Response

The response file is attached.
